# BELM: Bidirectional Explicit Linear Multi-step Sampler for Exact Inversion in Diffusion Models

**Fangyikang Wang**[1][*]   **Hubery Yin**[2][*]   **Yuejiang Dong**[3]   **Huminhao Zhu**[1]
**Chao Zhang**[1][†]   **Hanbin Zhao**[1]   **Hui Qian**[1]   **Chen Li**[2]
[1]Zhejiang University   [2]WeChat, Tencent Inc.   [3]Tsinghua University
{wangfangyikang,zhuhuminhao,zczju,zhaohanbin,qianhui}@zju.edu.cn
{hubery,chaselli}@tencent.com
dongyj21@mails.tsinghua.edu.cn

## Abstract

The inversion of diffusion model sampling, which aims to find the corresponding initial noise of a sample, plays a critical role in various tasks. Recently, several heuristic exact inversion samplers have been proposed to address the inexact inversion issue in a training-free manner. However, the theoretical properties of these heuristic samplers remain unknown and they often exhibit mediocre sampling quality. In this paper, we introduce a generic formulation, *Bidirectional Explicit Linear Multi-step* (BELM) samplers, of the exact inversion samplers, which includes all previously proposed heuristic exact inversion samplers as special cases. The BELM formulation is derived from the variable-stepsize-variable-formula linear multi-step method via integrating a bidirectional explicit constraint. We highlight this bidirectional explicit constraint is the key of mathematically exact inversion. We systematically investigate the Local Truncation Error (LTE) within the BELM framework and show that the existing heuristic designs of exact inversion samplers yield sub-optimal LTE. Consequently, we propose the Optimal BELM (O-BELM) sampler through the LTE minimization approach. We conduct additional analysis to substantiate the theoretical stability and global convergence property of the proposed optimal sampler. Comprehensive experiments demonstrate our O-BELM sampler establishes the exact inversion property while achieving high-quality sampling. Additional experiments in image editing and image interpolation highlight the extensive potential of applying O-BELM in varying applications.

## 1 Introduction

The emerging diffusion models (DMs) [52, 20, 55, 56], generating samples of data distribution from initial noise by learning a reverse diffusion process, have been proven to be an effective technique for modeling data distribution, especially in generating high-quality images [44, 10, 50, 46, 48, 21]. The diffusion process along with its sampling processes in DMs can be delineated as the forward and corresponding backward stochastic differential equations (SDE) [56, 1]. Furthermore, the sampling process can also be represented as a deterministic diffusion ordinary differential equation (ODE) [56, 53], which is also called Probability Flow ODE (PF-ODE) in some papers. Notably, the backward SDE and diffusion ODE share the same marginal distribution[56].

The inversion of the diffusion sampling, which aims to elucidate the correspondences between samples and initial noise, plays a critical role in various tasks of DMs. The diffusion inversion has

---

[*]Equal contribution. This work was done when Fangyikang Wang was an intern at WeChat.
[†]Corresponding author.

38th Conference on Neural Information Processing Systems (NeurIPS 2024).

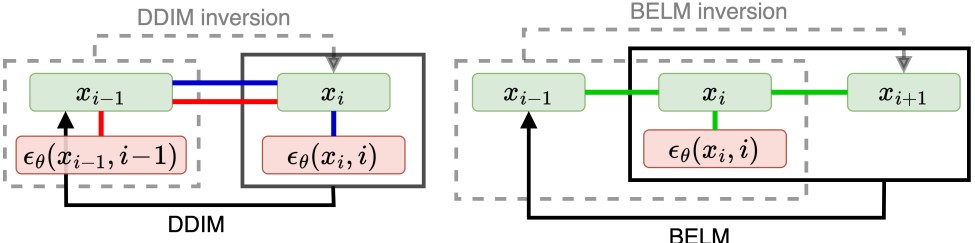

Figure 1: **Schematic description** of DDIM (left) and BELM (right). DDIM uses $\mathbf{x}_i$ and $\varepsilon_\theta(\mathbf{x}_i, i)$ to calculate $\mathbf{x}_{i-1}$ based on a linear relation between $\mathbf{x}_i$, $\mathbf{x}_{i-1}$ and $\varepsilon_\theta(\mathbf{x}_i, i)$ (represented by the blue line). However, DDIM inversion uses $\mathbf{x}_{i-1}$ and $\varepsilon_\theta(\mathbf{x}_{i-1}, i-1)$ to calculate $\mathbf{x}_i$ based on a different linear relation represented by the red line. This mismatch leads to the inexact inversion of DDIM. In contrast, BELM seeks to establish a linear relation between $\mathbf{x}_{i-1}$, $\mathbf{x}_i$, $\mathbf{x}_{i+1}$ and $\varepsilon_\theta(\mathbf{x}_i, i)$ (represented by the green line). BELM and its inversion are derived from this unitary relation, which facilitates the exact inversion. Specifically, BELM uses the linear combination of $\mathbf{x}_i$, $\mathbf{x}_{i+1}$ and $\varepsilon_\theta(\mathbf{x}_i, i)$ to calculate $\mathbf{x}_{i-1}$, and the BELM inversion uses the linear combination of $\mathbf{x}_{i-1}$, $\mathbf{x}_i$ and $\varepsilon_\theta(\mathbf{x}_i, i)$ to calculate $\mathbf{x}_{i+1}$. The bidirectional explicit constraint means this linear relation does not include the derivatives at the bidirectional endpoint, that is, $\varepsilon_\theta(\mathbf{x}_{i-1}, i-1)$ and $\varepsilon_\theta(\mathbf{x}_{i+1}, i+1)$.

a variety of downstream applications, including image editing [18, 57], image interpolation [53], inpainting [7], and super-resolution [67]. Several studies [31, 30, 7] have endeavored to tackle the inversion task within the context of SDE-based diffusion sampling. However, these works have not been able to achieve a mathematically exact inversion due to the inherent stochasticity of SDE.

In contrast, the diffusion ODE naturally gives out a correspondence between samples and noise. The famous DDIM [53] and its inversion are formulated by considering a first-order explicit Euler discretization to the diffusion ODE. However, as noted in the work of [18], the DDIM inversion introduces an inconsistency problem due to the schematic mismatch between DDIM and its inversion (see Figure 1). Encoding from $x_0$ to $x_T$ using DDIM inversion and then decoding using DDIM often leads to inexact reconstructions of the original samples (see Figure 4). To enable exact inversion, the work of null-text inversion [42] introduces intensive training for iterative optimization but still falls short of achieving a mathematically exact inversion.

Recently, several heuristic exact inversion samplers have been proposed to address this inexact inversion issue in a training-free manner [63, 71]. These samplers enable the mathematically exact inversion without the need for additional training and are thus compatible with pre-trained models. Taking inspiration from affine coupling layers in normalizing flows [11, 12], EDICT [63] intuitively introduces an auxiliary diffusion state and performs alternating mixture updates on the primal and auxiliary diffusion states. Later, BDIA [71] employs a symmetric bidirectional integration structure to achieve exact inversion intuitively. However, these heuristic exact inversion samplers often compromise the sampling quality due to their intuitive formula design. They may also introduce undesirable extra computational overhead or non-robust hyperparameters.

In this paper, we develop a generic formula for the general exact inversion samplers, termed as Bidirectional Explicit Linear Multi-step (BELM) samplers. We demonstrate that all previously proposed heuristic exact inversion samplers are, in fact, special instances of BELM samplers. The concept of BELM originates from the observation of the mismatch between DDIM formula and its inversion formula. BELM is formulated by establishing an unifying relationship, from which both BELM and its inversion are derived. More specifically, the unifying relationship of BELM is constructed in a variable-stepsize-variable-formula (VSVF) linear multi-step manner, supplemented with an additional bidirectional explicit constraint to facilitate exact inversion.

We systematically investigate the Local Truncation Error (LTE) within the BELM framework and show that the existing heuristic designs of exact inversion samplers yield sub-optimal LTE. Consequently, we employed a LTE minimization approach to design the formula of the optimal case within BELM, which we refer to as O-BELM. The formula for O-BELM dynamically adjusts in accordance with the timesteps, thereby ensuring minimized local error and consequently yielding the highest possible sampling accuracy. Furthermore, we demonstrate that O-BELM possesses the desirable property of zero-stability, which makes O-BELM robust to initial values. It also has the beneficial property of

global convergence, which prevents O-BELM from diverging during sampling. To the best of our knowledge, O-BELM is the first theoretically guaranteed exact inversion diffusion sampler.

We perform an image reconstruction experiment on the COCO dataset to validate that our O-BELM indeed achieves exact inversion, thereby enabling it to precisely recover complex image features. Furthermore, experiments involving both unconditional and conditional image generation demonstrate that O-BELM can ensure high sampling quality. Additional experiments in downstream tasks such as image editing and image interpolation highlight the extensive application potential of O-BELM.

## 2 Preliminaries

### 2.1 Diffusion Models and Diffusion SDEs

Suppose that we have a d-dimensional random variable $\mathbf{x}(0) \in \mathbb{R}^d$ following an unknown target distribution $q_0(x_0)$. Diffusion Models (DMs) define a forward process $\{\mathbf{x}(t)\}_{t \in [0,T]}$ with $T > 0$ starting with $\mathbf{x}(0)$, such that the distribution of $\mathbf{x}(t)$ conditioned on $\mathbf{x}(0)$ satisfies

$$q_{t|0}(\mathbf{x}(t)|\mathbf{x}(0)) = \mathcal{N}(\mathbf{x}(t); \alpha(t)\mathbf{x}(0), \sigma^2(t)\mathbf{I}), \tag{1}$$

where $\alpha(\cdot), \sigma(\cdot) \in \mathcal{C}([0,T], \mathbb{R}^+)$ have bounded derivatives, and we denote them as $\alpha_t$ and $\sigma_t$ for simplicity. The choice for $\alpha_t$ and $\sigma_t$ is referred to as the noise schedule of a DM. According to [33, 29, 38], with some assumption on $\alpha(\cdot)$ and $\sigma(\cdot)$, the forward process can be modeled as a linear SDE which is also called Ornstein–Uhlenbeck process:

$$d\mathbf{x}(t) = f(t)\mathbf{x}(t)dt + g(t)dB_t, \tag{2}$$

where $B_t$ is the standard d-dimensional Brownian Motion (BM), $f(t) = \frac{d \log \alpha_t}{dt}$ and $g^2(t) = \frac{d\sigma_t^2}{dt} - 2\frac{d \log \alpha_t}{dt}\sigma_t^2$. Under some regularity conditions, the above forward SDE (2) have a reverse SDE from time $T$ to 0, which starts from $\mathbf{x}(t)$ [1]:

$$d\mathbf{x}(t) = \left[f(t)\mathbf{x}(t) - g^2(t)\nabla_{\mathbf{x}(t)} \log q(\mathbf{x}(t), t)\right] dt + g(t)d\tilde{B}_t, \tag{3}$$

where $\tilde{B}_t$ is the reverse-time Brownian motion and $q(\mathbf{x}(t), t)$ is the single-time marginal distribution of the forward process. In practice, DMs [20, 56] use $\varepsilon_\theta(\mathbf{x}(t), t)$ to estimate $-\sigma(t)\nabla_{\mathbf{x}(t)} \log q(\mathbf{x}(t), t)$ and the parameter $\theta$ is optimized by the following objective:

$$\theta^* = \arg\min_\theta \mathbb{E}_t \left\{\lambda_t \mathbb{E}_{x_0, x_t} \left[\|s_\theta(x_t, t) - \nabla_{x_t} \log p(x_t, t|x_0, 0)\|^2\right]\right\}, \tag{4}$$

### 2.2 Diffusion ODE and DDIM

It is noted that the reverse SDE (3) has an associated probability flow ODE (also called diffusion ODE), which is a deterministic process that shares the same single-time marginal distribution [56]:

$$d\mathbf{x}(t) = \left[f(t)\mathbf{x}(t) - \frac{1}{2}g^2(t)\nabla_{\mathbf{x}(t)} \log q(\mathbf{x}(t), t)\right] dt. \tag{5}$$

Upon substituting the $f(t)$ and $g(t)$ into Eq. (5), we obtain the following first-order form:

$$d\left(\frac{\mathbf{x}(t)}{\alpha_t}\right) = \varepsilon_\theta(\mathbf{x}(t), t) d\left(\frac{\sigma_t}{\alpha_t}\right). \tag{6}$$

The famous DDIM sampler [53] can be obtained by applying the explicit Euler method to Eq. (6).

$$\mathbf{x}_{i-1} = \frac{\alpha_{i-1}}{\alpha_i}\mathbf{x}_i + \left(\sigma_{i-1} - \frac{\alpha_{i-1}}{\alpha_i}\sigma_i\right)\varepsilon_\theta(\mathbf{x}_i, i). \tag{7}$$

The inversion of DDIM is obtained by applying the explicit Euler method in the reverse of Eq. (6):

$$\mathbf{x}_i = \frac{\alpha_i}{\alpha_{i-1}}\mathbf{x}_{i-1} + \left(\sigma_i - \frac{\alpha_i}{\alpha_{i-1}}\sigma_{i-1}\right)\varepsilon_\theta(\mathbf{x}_{i-1}, i-1). \tag{8}$$

## 2.3 Intuitive Exact Inversion Samplers of Diffusion Models

In practice, we observe an inconsistency issue with the DDIM inversion (8). Consider a sample $\mathbf{x}_0$; using DDIM inversion, we obtain the corresponding noise $\mathbf{x}_T$ and then use DDIM to reconstruct a $\mathbf{x}_0^*$. The reconstructed $\mathbf{x}_0^*$ would exhibit significant inconsistency with the original sample $\mathbf{x}_0$. Recently, two exact inversion samplers, EDICT and BDIA, have been heuristically proposed to address this inconsistency issue in a training-free manner.

**EDICT sampler**  Taking inspiration from affine coupling layers in normalizing flows [11, 12], the recent work [63] proposed EDICT to enforce exact diffusion inversion. The basic idea lies in introducing an auxiliary diffusion state $\mathbf{y}_t$ to be coupled with $\mathbf{x}_t$. Denoting $a_i = \frac{\alpha_{i-1}}{\alpha_i}$ and $b_i = \sigma_{i-1} - \frac{\alpha_{i-1}}{\alpha_i}\sigma_i$, the formulation of EDICT writes:

$$
\begin{cases}
\mathbf{x}_i^{inter} = a_i\mathbf{x}_i + b_i\boldsymbol{\varepsilon}_\theta(\mathbf{y}_i, i), & \mathbf{y}_i^{inter} = a_i\mathbf{y}_i + b_i\boldsymbol{\varepsilon}_\theta(\mathbf{x}(t)^{inter}, i), \\
\mathbf{x}_{i-1} = p\mathbf{x}_i^{inter} + (1-p)\mathbf{y}_i^{inter}, & \mathbf{y}_{i-1} = p\mathbf{y}_i^{inter} + (1-p)\mathbf{x}_{i-1}.
\end{cases}
\tag{9}
$$

where $p \in (0, 1)$ is the mixing coefficient. The details of EDICT inversion defers to Appendix A.1.

**BDIA sampler**  BDIA sampler [71] utilizes a symmetric bidirectional integration structure to achieve exact inversion. BDIA reformulate the expression of DDIM (7) to be $\mathbf{x}_{i-1}^{\text{DDIM}} = \mathbf{x}_i^{\text{DDIM}} + \Delta\left(i \to i-1 | \mathbf{x}_i^{\text{DDIM}}\right)$ and the expression of DDIM inversion (8) to be $\mathbf{x}_i^{\text{DDIM}} = \mathbf{x}_{i-1}^{\text{DDIM}} + \Delta\left(i-1 \to i | \mathbf{x}_{i-1}^{\text{DDIM}}\right)$. BDIA intuitively leverage $-\left[(1-\gamma)(\mathbf{x}_{i+1} - \mathbf{x}_i) + \gamma\Delta\left(i \to i+1 | \mathbf{x}_i\right)\right]$ to approximate the increment from $x_{i+1}$ to $x_i$ and $\Delta\left(i \to i-1 | \mathbf{x}_i\right)$ as the increment from $x_i$ to $x_{i-1}$. Thus, the updating rule of BDIA writes:

$$
\mathbf{x}_{i-1} = \mathbf{x}_{i+1} \underbrace{- \left[(1-\gamma)(\mathbf{x}_{i+1} - \mathbf{x}_i) + \gamma\Delta\left(i \to i+1 | \mathbf{x}_i\right)\right]}_{increment(\mathbf{x}_{i+1} \to \mathbf{x}_i)} + \underbrace{\Delta\left(i \to i-1 | \mathbf{x}_i\right)}_{increment(\mathbf{x}_i \to \mathbf{x}_{i-1})}.
\tag{10}
$$

The comprehensive formulation of BDIA and its inversion can be found in Appendix A.2.

However, the theoretical properties of these heuristic samplers remain unknown and they often exhibit compromised sampling quality. To the best of our knowledge, there is no systematic approach to derive a diffusion sampler that simultaneously possesses the exact diffusion inversion property and maintains high sampling quality.

## 3 The Generic Bidirectional Explicit Linear Multi-step (BELM) Samplers

In this section, we first model the diffusion sampling process as a well-posed initial value problem to facilitate subsequent analysis. By the rethinking of DDIM inversion, we propose the generic Bidirectional Explicit Linear Multi-step (BELM) samplers in a variable-stepsize-variable-formula (VSVF) manner. We further illustrate that EDICT and BDIA are, in fact, special instances of the BELM framework.

**The diffusion sampling problem as an IVP**  By denoting $\bar{\mathbf{x}}(t) \equiv \frac{\mathbf{x}(t)}{\alpha_t}$, $\bar{\sigma}(t) \equiv \frac{\sigma_t}{\alpha_t}$ and $\bar{\boldsymbol{\varepsilon}}_\theta(\bar{\mathbf{x}}(t), \bar{\sigma}_t) \equiv \boldsymbol{\varepsilon}_\theta(\mathbf{x}(t), t)$, the deterministic sampling process of DMs (6) can be seen as an special reverse-time diffusion initial value problem (IVP) [58, p.310][3, p.3]:

$$
d\bar{\mathbf{x}}(t) = \bar{\boldsymbol{\varepsilon}}_\theta\left(\bar{\mathbf{x}}(t), \bar{\sigma}_t\right) d\bar{\sigma}_t,
\tag{11}
$$

where $\bar{\mathbf{x}}(T) = \mathbf{x}(T)/\alpha_T$. A fundamental question before any further analysis is whether the given diffusion IVP (11) admits any solution and, if so, whether this solution is unique. Firstly, we need to establish some regularity assumptions on our diffusion sampling problem (6).

> **Assumption 1.** $\varepsilon_\theta(\mathbf{x}, t)$ *is continuous w.r.t.* $t$ *and Lipschitz continuous w.r.t.* $\mathbf{x}$ *with the Lipschitz constant* $L_{\boldsymbol{\varepsilon}_\theta}$, *which implies* $\|\varepsilon_\theta(\mathbf{x}_1, t) - \varepsilon_\theta(\mathbf{x}_2, t)\|_2 \leq L_{\boldsymbol{\varepsilon}_\theta}\|\mathbf{x}_1 - \mathbf{x}_2\|_2$.

The Assumption 1 is a common assumption of the noise predictor $\varepsilon_\theta(\mathbf{x}, t)$ in the DMs literature [54]. Under the condition of Assumption 1, we can confirm the diffusion IVP (11) is well-posed by a direct application of the existence and uniqueness theorem in the IVP theory [3, p. 23].

> **Proposition 1.** *Under Assumption 1, there exists a unique solution to the diffusion IVP* (11).

In this paper, $\mathbf{x}(\cdot)$ denote the continuous solution, and $\mathbf{x}_i$ denote numerical approximations.

**Rethinking on DDIM inversion** As shown in Figure 1, DDIM (7) and its inversion (8) are derived based on different linear relationships. We highlight that this mismatch results in the inexact inversion of DDIM. Building on this observation, a natural idea is to construct the DDIM inversion based on the same linear relationships as the DDIM to eliminate this mismatch. Regrettably, DDIM is constructed on a relationship between $\mathbf{x}_i$, $\mathbf{x}_{i-1}$, and $\varepsilon_\theta(\mathbf{x}_i, i)$ (utilizes $\mathbf{x}_i$, and $\varepsilon_\theta(\mathbf{x}_i, i)$ to compute $\mathbf{x}_{i-1}$), which DDIM inversion cannot leverage to directly calculate $\mathbf{x}_i$, as $\varepsilon_\theta(\mathbf{x}_i, i)$ is also unknown in the DDIM inversion case. This relation is explicit for DDIM but implicit for DDIM inversion. It should be noted that implicit equations must be solved using iterative methods such as Newton's method [58, p. 19], which are time-consuming and can introduce numerical error in the context of DMs [23, 39].

To address this issue, we establish a new relationship between adjacent states and derivatives, which can be explicitly computed in both directions. Subsequently, we formulate both the sampler and its inversion based on this singular linear relationship to achieve exact inversion. This is the fundamental concept of BELM samplers.

**Bidirectional Explicit Linear Multi-step (BELM) samplers** In an attempt to establish a linear relationship between $\mathbf{x}_i$, $\mathbf{x}_{i-1}$, $\varepsilon_\theta(\mathbf{x}_i, i)$, and $\varepsilon_\theta(\mathbf{x}_{i-1}, i-1)$ that can be explicitly computed bidirectionally, we must exclude both $\varepsilon_\theta(\mathbf{x}_i, i)$ and $\varepsilon_\theta(\mathbf{x}_{i-1}, i-1)$. However, this exclusion results in a relationship that lacks sufficient information. Consequently, it becomes imperative to take more states into account. This prompts us to explore the concept of the linear multi-step (LM) method [3, p.111] as a means to derive a linear relationship between adjacent states and the derivatives of the diffusion IVP. However, the commonly used noise schedule of DMs would lead to a non-equidistant series of $\{\bar{\sigma}_i\}, i = 1 \ldots N$. So, instead of the classical LM methods with fixed stepsize, we shall consider it in the variable-stepsize-variable-formula (VSVF) manner [8], which use dynamic multistep formulae w.r.t. different stepsizes. Let $t_0 < t_1 < \ldots t_N = t_0 + T$ be a grid in $[t_0, t_0 + T]$, $h_i = \bar{\sigma}_i - \bar{\sigma}_{i-1}, i = N \ldots 1, h_0 = \bar{\sigma}_0$ and $h = \max h_i$, the k-step VSVF LM methods w.r.t. Eq. (11) will calculate $\bar{\mathbf{x}}_{i-1}$ at the points $\bar{\sigma}_{i-1}$ with the following difference equation:

$$\bar{\mathbf{x}}_{i-1} = \sum_{j=1}^{k} a_{i,j} \cdot \bar{\mathbf{x}}_{i-1+j} + \sum_{j=0}^{k} b_{i,j} \cdot h_{i-1+j} \cdot \bar{\varepsilon}_\theta(\bar{\mathbf{x}}_{i-1+j}, \bar{\sigma}_{i-1+j}), \tag{12}$$

where the coefficient of updates and stepsizes are all dependent on $i$. Throughout this paper, any reference to LM will, by default, imply VSVF LM unless explicitly stated otherwise. If $b_{i,0} = 0$ for all $i$ in Eq. (12), the method is called **explicit**, since the formula can directly compute $\bar{\mathbf{x}}_{i-1}$. Clearly, the LM (12) have a reversed formula which is also a k-step LM as follows (assume $a_{i,k} \neq 0$),

$$\bar{\mathbf{x}}_{i-1+k} = \frac{1}{a_{i,k}} \cdot \bar{\mathbf{x}}_{i-1} - \sum_{j=1}^{k-1} \frac{a_{i,j}}{a_{i,k}} \cdot \bar{\mathbf{x}}_{i-1+j} + \sum_{j=0}^{k} \frac{b_{i,j}}{a_{i,k}} \cdot h_{i-1+j} \cdot \bar{\varepsilon}_\theta(\bar{\mathbf{x}}_{i-1+j}, \bar{\sigma}_{i-1+j}). \tag{13}$$

If the reversed VSVFM is explicit, i.e. $b_{i,k} = 0$ for all $i$, we call the origin LM (12) to be **backward explicit**. Now we can define a k-step LM to be **bidirectional explicit** when it is explicit as well as backward explicit. We call the LM samplers abide by the bidirectional explicit constraint as the Bidirectional Explicit Linear Multi-step (**BELM**) samplers, which have the general form:

$$\bar{\mathbf{x}}_{i-1} = \sum_{j=1}^{k} a_{i,j} \cdot \bar{\mathbf{x}}_{i-1+j} + \sum_{j=1}^{k-1} b_{i,j} \cdot h_{i-1+j} \cdot \bar{\varepsilon}_\theta(\bar{\mathbf{x}}_{i-1+j}, \bar{\sigma}_{i-1+j}). \tag{14}$$

We highlight this bidirectional explicit constraint is key to mathematically exact diffusion inversion:

> **Proposition 2.** *Any BELM method* (14) *with* $a_{i,k} \neq 0$ *has the exact inversion property.*

Table 1: Theoretical properties comparison of different samplers.

| | exact inversion | local error | zero-stable | global convergence |
|---|---|---|---|---|
| | | Theoretical properties | | |
| DDIM[53] | ✗ | $\mathcal{O}\left(\alpha_i h_i{}^2\right)$ | ✓ | ✓ |
| EDICT[63] | ✓ | $\mathcal{O}\left(\sqrt{\alpha_{i-1}} h_i\right)$ | unclear | unclear |
| BDIA[71] | ✓ | $\mathcal{O}\left(\alpha_i (h_i + h_{i+1})^2\right)$ | unclear | unclear |
| **O-BELM (Ours)** | ✓ | $\mathcal{O}\left(\alpha_i (h_i + h_{i+1})^3\right)$ | ✓ | ✓ |

As an instance, setting $k = 2$ in Eq. (14) yields the 2-step BELM diffusion sampler:

$$\bar{\mathbf{x}}_{i-1} = a_{i,2}\bar{\mathbf{x}}_{i+1} + a_{i,1}\bar{\mathbf{x}}_i + b_{i,1}h_i\bar{\boldsymbol{\varepsilon}}_\theta(\bar{\mathbf{x}}_i, \bar{\sigma}_i). \tag{15}$$

For detailed information on the 3-step BELM diffusion sampler, the general k-step case, and their optimal design, readers are referred to Appendix A.4 and A.5. In the main body of this paper, we will default mean 2-step case unless explicitly stated.

**BDIA and EDICT as special case of BELM** We find that, although developed from heuristic ideas, both BDIA and EDICT are special cases within the BELM framework. That is, their exact inversion property is inherited from the fact that they are fundamentally instances of BELM samplers.

> **Remark 1.** *EDICT* (9) *and BDIA* (10) *are both special cases within the BELM framework.*

The detailed mathematical derivation for Remark 1 can be found in Appendices A.7 and A.8.

## 4 The Optimal-BELM (O-BELM) Sampler

In this section, we systematically investigate the Local Truncation Error (LTE) within the BELM framework and show that the existing heuristic designs of exact inversion samplers yield sub-optimal LTE. Consequently, we introduce Optimal-BELM (O-BELM), which utilizes a more refined dynamic formula developed through the LTE minimization approach. Additional analysis is conducted to substantiate the theoretical stability and global convergence property of O-BELM.

### 4.1 Analysis on Local Truncation Error

The Local Truncation Error (LTE) quantifies the error introduced in a step update. Specifically, it computes the difference between the numerical solution and its underlying true solution, assuming perfect knowledge of the true solution at the previous states.

> **Definition 1.** *The LTE of BELM* (15) *on $\bar{\mathbf{x}}_i$ at each step $i$ is defined as :*
> $$\tau_i = \bar{\mathbf{x}}(t_{i-1}) - a_{i,2}\bar{\mathbf{x}}_{i+1} - a_{i,1}\bar{\mathbf{x}}_i - b_{i,1}h_i\bar{\boldsymbol{\varepsilon}}_\theta(\bar{\mathbf{x}}_i, \bar{\sigma}_i). \tag{16}$$

Under Assumption 2 (details in Appendix A.3), we can utilize the Taylor expansion to investigate the LTE of BELM (15) as follows:

> **Proposition 3.** *Under Assumption 2, the LTE of the BELM* (15) *gives general form as follows:*
> $$\tau_i = c_{i,1}\bar{\mathbf{x}}(t_{i-1}) + c_{i,2}\bar{\boldsymbol{\varepsilon}}_\theta\left(\bar{\mathbf{x}}(t_{i-1}), \bar{\sigma}_{i-1}\right) + c_{i,3}\nabla_{\bar{\sigma}_{i-1}}\bar{\boldsymbol{\varepsilon}}_\theta\left(\bar{\mathbf{x}}(t_{i-1}), \bar{\sigma}_{i-1}\right) + \mathcal{O}\left((h_i + h_{i+1})^3\right),$$
> $$\tag{17}$$
> *where $c_{i,1} = 1 - a_{i,1} - a_{i,2}$, $c_{i,2} = -a_{i,1}h_i - a_{i,2}(h_i + h_{i+1}) - b_{i,1}h_i$, and $c_{i,3} = -\frac{a_{i,1}}{2}h_i^2 - \frac{a_{i,2}}{2}(h_i + h_{i+1})^2 - b_{i,1}h_i^2$.*

In the task of DMs, our primary concern is the LTE on $\mathbf{x}_{i-1}$ rather than $\bar{\mathbf{x}}_{i-1}$. We denote the LTE on $\mathbf{x}_i$ as $\mathbf{e}_i$. It is clear that $\mathbf{e}_i = \alpha_{i-1}\tau_i$. We investigate the LTE of existing samplers as follows:

**Corollary 1.** *Under Assumption 2, the LTE $\mathbf{e}_i$ of DDIM sampler* (7) *is $\mathcal{O}\left(\alpha_{i-1}h_i{}^2\right)$; The LTE $\mathbf{e}_i$ of BDIA sampler* (10) *is $\mathcal{O}\left(\alpha_{i-1}(h_i + h_{i+1})^2\right)$ for any fixed $\gamma \in [0,1]$; The LTE $\mathbf{e}_i$ of EDICT sampler* (9) *is $\mathcal{O}\left(\sqrt{\alpha_{i-1}}h_i\right)$ for any constant $p \in (0,1)$.*

## 4.2 Optimal BELM Sampler via LTE Minimization

We then demonstrate that, through a meticulous design of formulae, we can achieve a higher order of LTE within the BELM framework compared to existing sub-optimal instances. Specifically, we utilize an LTE minimization approach, inspired by the design of renowned LM methods such as the Adams–Bashforth methods [2] or the Adams–Moulton methods [43, 40].

**Proposition 4.** *Under Assumption 2, the LTE $\tau_i$ of BELM diffusion sampler* (15) *can be accurate up to $\mathcal{O}\left((h_i + h_{i+1})^3\right)$ when formulae are designed as $a_{i,1} = \frac{h_{i+1}^2 - h_i^2}{h_{i+1}^2}, a_{i,2} = \frac{h_i^2}{h_{i+1}^2}, b_{i,1} = -\frac{h_i + h_{i+1}}{h_{i+1}}$.*

When this is satisfied, obviously, the LTE $\mathbf{e}_i$ on $\mathbf{x}_{i-1}$ is $\mathcal{O}\left(\alpha_{i-1}(h_i + h_{i+1})^3\right)$. Substituting the designed formulas into (15), we derive the Optimal-BELM (O-BELM) sampler:

$$\mathbf{x}_{i-1} = \frac{h_i^2}{h_{i+1}^2}\frac{\alpha_{i-1}}{\alpha_{i+1}}\mathbf{x}_{i+1} + \frac{h_{i+1}^2 - h_i^2}{h_{i+1}^2}\frac{\alpha_{i-1}}{\alpha_i}\mathbf{x}_i - \frac{h_i(h_i + h_{i+1})}{h_{i+1}}\alpha_{i-1}\varepsilon_\theta(\mathbf{x}_i, i). \tag{18}$$

The inversion of O-BELM diffusion sampler (18) writes:

$$\mathbf{x}_{i+1} = \frac{h_{i+1}^2}{h_i^2}\frac{\alpha_{i+1}}{\alpha_{i-1}}\mathbf{x}_{i-1} + \frac{h_i^2 - h_{i+1}^2}{h_i^2}\frac{\alpha_{i+1}}{\alpha_i}\mathbf{x}_i + \frac{h_{i+1}(h_i + h_{i+1})}{h_i}\alpha_{i+1}\varepsilon_\theta(\mathbf{x}_i, i). \tag{19}$$

## 4.3 Further Theoretical Analysis on O-BELM

Here, we further demonstrate that the O-BELM not only surpasses in terms of local accuracy but also excels in **stability** and **global convergence** properties.

As is clear from (15), we need starting values before we can apply a method to the diffusion IVP. Of these, the initial one is given by the initial condition, but the others, have to be computed by other means, say, by using DDIM. At any rate, the starting values will contain numerical errors and it is crucial to ensure that perturbations of the initial values do not lead to an error explosion in the subsequent steps. This concept is encapsulated in numerical analysis as zero-stability.

**Definition 2.** *The LM* (12) *is said to be **zero-stable** if there exists a constant $K$ such that, for any two sequences $\{\bar{\mathbf{x}}_i\}$ and $\{\bar{\mathbf{z}}_i\}$ that have been generated by the same formulae but different starting values $\bar{\mathbf{x}}_N, \bar{\mathbf{x}}_{N-1}, \ldots, \bar{\mathbf{x}}_{N-k+1}$ and $\bar{\mathbf{z}}_N, \bar{\mathbf{z}}_{N-1}, \ldots, \bar{\mathbf{z}}_{N-k+1}$, respectively, we have*

$$\|\bar{\mathbf{x}}_i - \bar{\mathbf{z}}_i\| \le K \max\left\{\|\bar{\mathbf{x}}_N - \bar{\mathbf{z}}_N\|, \|\bar{\mathbf{x}}_{N-1} - \bar{\mathbf{z}}_{N-1}\|, \ldots, \|\bar{\mathbf{x}}_{N-k+1} - \bar{\mathbf{z}}_{N-k+1}\|\right\}, \tag{20}$$

*for all $i$, and as $h$ tends to 0.*

We also want to ensure that a method will gradually converge to the underlying truth as the stepsizes decrease, a concept that aligns with the global convergence property.

**Definition 3.** *The LM* (12) *is **globally convergent** if for every solution $\bar{\mathbf{x}}(t)$ of* (11)

$$\lim_{h \to 0}\max_{0 \le i \le N}\|\bar{\mathbf{x}}_i - \bar{\mathbf{x}}(t_i)\| = 0, \tag{21}$$

*when initial error $\sum_{j=N}^{N-1+k}\left(\|\bar{\mathbf{x}}_j - \bar{\mathbf{x}}(t_j)\| + h_i\|\bar{\varepsilon}_\theta(\bar{\mathbf{x}}_j, \bar{\sigma}_j) - \bar{\varepsilon}_\theta(\bar{\mathbf{x}}(t_j), \bar{\sigma}_j)\|\right)$ tends to zero.*

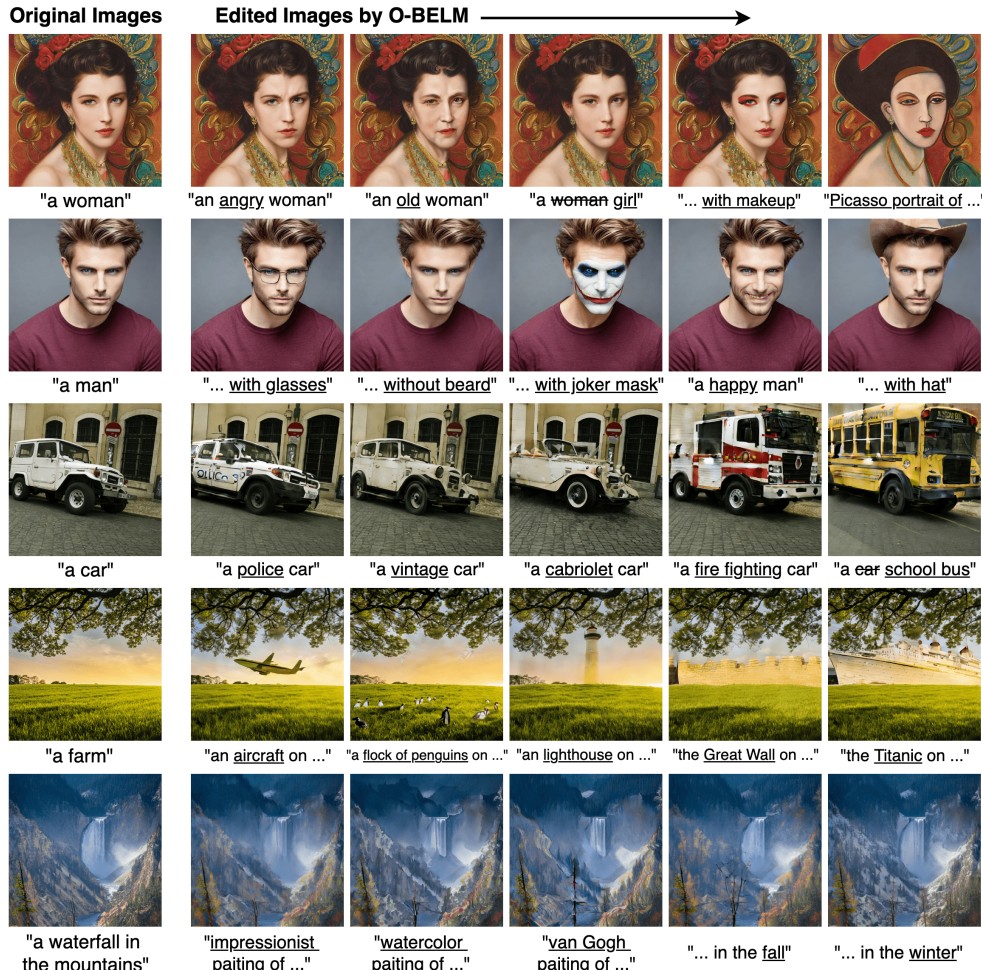

**Original Images**    **Edited Images by O-BELM** ⟶

"a woman"   "an angry woman"   "an old woman"   "a ~~woman~~ girl"   "... with makeup"   "Picasso portrait of ..."

"a man"   "... with glasses"   "... without beard"   "... with joker mask"   "a happy man"   "... with hat"

"a car"   "a police car"   "a vintage car"   "a cabriolet car"   "a fire fighting car"   "a ~~car~~ school bus"

"a farm"   "an aircraft on ..."   "a flock of penguins on ..."   "an lighthouse on ..."   "the Great Wall on ..."   "the Titanic on ..."

"a waterfall in the mountains"   "impressionist paiting of ..."   "watercolor paiting of ..."   "van Gogh paiting of ..."   "... in the fall"   "... in the winter"

Figure 2: Examples of editing results using O-BELM on both synthesized and real images. We showcase the diverse editing capabilities of O-BELM across a range of tasks, including human face modifications, content change, entity addition and global style transfer. The exact inversion property of O-BELM enables large-scale image alterations while preserving auxiliary details (background in first row, hairstyle in second row, traffic sign in third row, tree and crop in fourth row, composition in last row). Its stability and accuracy further ensure the high quality of the resulting images.

We affirm that our O-BELM sampler possesses the nice zero-stable property as well as the global convergence property.

**Proposition 5.** *The O-BELM sampler* (18) *is (a)* ***zero-stable*** *and (b)* ***globally convergent***.

## 5   Experiments

In this section, we conduct experiments to verify that O-BELM achieves the exact inversion property while maintaining high-quality sampling ability. We further demonstrate the extensive potential of applying the O-BELM sampler in various applications, such as image editing and image interpolation (deferred to Appendix C.3). All the pre-trained models utilized are listed in Appendix C.5.

### 5.1   Image Reconstruction

We adopt the experimental setting from [63] to demonstrate the exact diffusion inversion property of O-BELM using 10k images in the MS-COCO-2014 validation set [35]. Given an image, inverted latents are calculated and used to reconstruct the image using SD-1.5. Mean-square error (MSE) is calculated on pixels normalized to $[-1, 1]$ and averaged across 10k images. The autoencoder (AE)

Table 2: Comparison of different samplers on MSE reconstruction loss on COCO-14.

|  | MSE loss of reconstruction | | | | |
|---|---|---|---|---|---|
|  | DDIM | AE | EDICT | BDIA | **O-BELM** |
| 10 steps | 0.026 | 0.004 | 0.004 | 0.004 | 0.004 |
| 20 steps | 0.016 | 0.004 | 0.004 | 0.004 | 0.004 |
| 50 steps | 0.008 | 0.004 | 0.004 | 0.004 | 0.004 |
| 100 steps | 0.007 | 0.004 | 0.004 | 0.004 | 0.004 |

Table 3: Comparison of different samplers on FID score( ↓) for the task of unconditional generation.

|  | CIFAR10 (32 × 32) | | | | CelebA-HQ (256 × 256) | | | |
|---|---|---|---|---|---|---|---|---|
|  | DDIM | EDICT | BDIA | **O-BELM** | DDIM | EDICT | BDIA | **O-BELM** |
| 10 steps | 17.45 | 87.11 | 12.27 | **10.98** | 27.13 | 57.82 | 27.41 | **19.13** |
| 20 steps | 10.60 | 38.84 | **7.27** | **7.17** | 16.33 | 39.24 | 16.18 | **11.54** |
| 50 steps | 6.96 | 10.24 | 5.77 | **5.24** | **10.77** | 16.72 | **10.65** | **10.41** |
| 100 steps | 5.72 | 5.31 | 5.07 | **4.18** | **10.19** | 12.24 | **10.30** | **10.17** |

reconstruction error in the SD pipeline serves as a lower bound. From Table 2, we observe that, regardless of the stepsize, O-BELM and its sub-optimal siblings BDIA and EDICT consistently achieve the lowest MSE, signifying their exact inversion at the latent level. In contrast, DDIM tends to suffer from inconsistency. More visual reconstruction examples can be found in Appendix C.1.

## 5.2 Unconditional Image Generation

In this section, we conduct an unconditional image generation task to validate the high-quality sampling ability of O-BELM. Utilizing a pre-trained model, we generate 50k artificial images over a specific number of steps and compute the corresponding Fréchet Inception Distance (FID) score with the real data. Specifically, Fréchet Inception Distance (FID) [19] calculates the Fréchet distance between the real data and the generated data. A lower FID implies more realistic generated data. Table 3 summarizes the computed FID scores for the CIFAR10 and CelebA-HQ datasets. It is evident that O-BELM consistently outperforms other exact inversion samplers in terms of sampling quality. This experimental result corroborates the error analysis presented in Table 1. The parameters $\gamma$ for BDIA and $p$ for EDICT are determined through grid search. Details can be found in Appendix C.2.

## 5.3 Conditional Image Generation

We further evaluate these samplers under conditional image generation tasks. We employ the StableDiffusion V1.5 and V2-base models to generate 30k images of resolution 512×512, based on text prompts from the COCO-14 validation set. All methods utilize the same seed and the same text prompts set. As evident from Table 4, O-BELM also exhibits superior sampling quality in the context of conditional image generation. We ensure a fair comparison by selecting appropriate guidance weights and hyperparameters, details of which can be found in Appendix C.2.

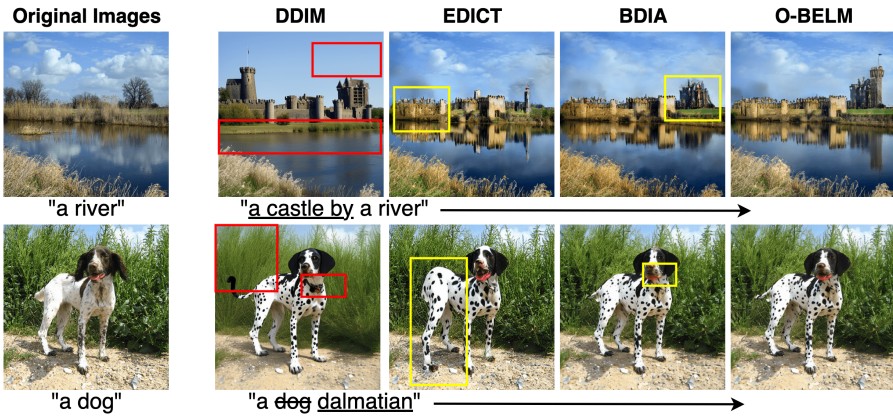

Figure 3: Comparison of editing results from different samplers under 50 steps. DDIM leads to inconsistencies (highlighted by the red rectangle), and the EDICT and BDIA samplers may introduce unrealistically low-quality sections (highlighted by the yellow rectangle). Our O-BELM sampler ensures consistency and demonstrates high-quality results.

Table 4: Comparison of different samplers on FID score( ↓) for the task of text-to-image generation with pretrained stable diffusion models.

| | SD-1.5 (512 × 512) | | | | SD-2.0-base (512 × 512) | | | |
|---|---|---|---|---|---|---|---|---|
| | DDIM | EDICT | BDIA | **O-BELM** | DDIM | EDICT | BDIA | **O-BELM** |
| 10 steps | 21.44 | 85.77 | 23.96 | **18.19** | 20.40 | 75.14 | 22.00 | **17.01** |
| 20 steps | 19.45 | 27.17 | 20.39 | **17.92** | 18.57 | 24.15 | 18.72 | **16.53** |
| 50 steps | 18.93 | 21.30 | 19.38 | **17.96** | 17.82 | 19.76 | 17.98 | **16.52** |
| 100 steps | 18.83 | 21.13 | 19.21 | **18.19** | 17.64 | 19.49 | 17.86 | **16.75** |

## 5.4 Training-free Image Editing

In this section, we present the results of the O-BELM sampler in an image editing task as shown in Figure 2, and compare the editing effects of different samplers in Figure 3. We demonstrate that the exact inversion property of O-BELM ensures the preservation of image features that we do not wish to edit. Furthermore, we illustrate how the high accuracy and stability properties of O-BELM contribute to the high quality of the edited image.

We emphasis that the goal of experiments here is not going to use our O-BELM sampler alone to achieve commercial-grade level image editing. It's quite unfair for training-free exact sampler methods to compete with commercial-grade image editing pipelines involving domain-specific training [25, 68], attention modification [18, 45], testing-time finetuning [62, 24, 6], complex control [73], real-data inversion alignment [75] or input text refinement [47, 37, 32]. In fact, our O-BELM sampler is orthogonal to these image editing techniques, using a better exact inversion sampler like O-BELM in the commercial-grade image editing pipeline remains a promising future work.

## 6 Conclusions

We tackle the inexact inversion issue of DMs in a training-free manner. We introduce the generic Bidirectional Explicit Linear Multi-step (BELM) framework based on a linear multi-step observation, which encompasses existing heuristic exact inversion samplers as special cases. Furthermore, we devise a Local Truncation Error (LTE) minimization approach to construct the Optimal-BELM (O-BELM) within the BELM framework, which achieves a higher order of local error. We provide a theoretical guarantee of global stability and convergence for O-BELM and conduct various experiments to demonstrate that O-BELM not only accomplishes exact inversion but also maintains a high-quality sampling capability. Please refer to further discussion and limitations in appendix D. The code repository can be found at `https://github.com/zituitui/BELM`.

## 7 Acknowledgments

This work was supported in part by National Natural Science Foundation of China under Grant 62206248 and National Natural Science Foundation of China under Grant 62402430. We would like to thank all the reviewers for their constructive comments. Fangyikang Wang wishes to express gratitude to Pengze Zhang from ByteDance, as well as Yiling Zhang and Yinan Li from Zhejiang University, for their insightful discussions on the experiments.

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

# Appendix

## Contents

# A    Formulations

## A.1    Detail Formulation of EDICT

A sequential inversion and rearrangement of EDICT (9) yields the EDICT inversion:

$$\begin{cases} \mathbf{y}_i^{inter} = (\mathbf{y}_{i-1} - (1-p)\mathbf{x}_{i-1})/p, \\ \mathbf{x}_i^{inter} = (\mathbf{x}_{i-1} - (1-p)\mathbf{y}_i^{inter})/p, \\ \mathbf{y}_i = \left( \mathbf{y}_i^{inter} - b_i \varepsilon_\theta(\mathbf{x}_i^{inter}, i) \right)/a_i, \\ \mathbf{x}_i = \left( \mathbf{x}_i^{inter} - b_i \varepsilon_\theta(\mathbf{y}_i, i) \right)/a_i. \end{cases} \tag{22}$$

## A.2    Detail Formulation of BDIA

BDIA sampler [71] utilizes bi-directional integration to achieve exact inversion, also introducing an additional hyperparameter. Reformulate the expression of DDIM (7) to be $\mathbf{x}_{i-1}^{\mathrm{DDIM}} = \mathbf{x}_i^{\mathrm{DDIM}} + \Delta\left(i \to i-1|\mathbf{x}_i^{\mathrm{DDIM}}\right)$ and the expression of DDIM inversion (8) to be $\mathbf{x}_i^{\mathrm{DDIM}} = \mathbf{x}_{i-1}^{\mathrm{DDIM}} + \Delta\left(i-1 \to i|\mathbf{x}_{i-1}^{\mathrm{DDIM}}\right)$, that is,

$$\begin{cases} \Delta\left(i \to i-1|\mathbf{x}_i\right) = \left( \dfrac{\alpha_{i-1}}{\alpha_i} - 1 \right)\mathbf{x}_i + \left( \sigma_{i-1} - \dfrac{\alpha_{i-1}}{\alpha_i}\sigma_i \right)\varepsilon_\theta(\mathbf{x}_i, i) \\ \Delta\left(i-1 \to i|\mathbf{x}_{i-1}\right) = \left( \dfrac{\alpha_i}{\alpha_{i-1}} - 1 \right)\mathbf{x}_{i-1} + \left( \sigma_i - \dfrac{\alpha_i}{\alpha_{i-1}}\sigma_{i-1} \right)\varepsilon_\theta(x_{i-1}, i-1). \end{cases} \tag{23}$$

The updating rule of BDIA write:

$$\begin{aligned} \mathbf{x}_{i-1} =& \mathbf{x}_{i+1} \underbrace{- \left[ (1-\gamma)(\mathbf{x}_{i+1} - \mathbf{x}_i) + \gamma\Delta\left(i \to i+1|\mathbf{x}_i\right) \right]}_{increment(\mathbf{x}_{i+1} \to \mathbf{x}_i)} + \underbrace{\Delta\left(i \to i-1|\mathbf{x}_i\right)}_{increment(\mathbf{x}_i \to \mathbf{x}_{i-1})}, \\ =& \mathbf{x}_{i+1} - (1-\gamma)(\mathbf{x}_{i+1} - \mathbf{x}_i) - \gamma\left[ \left( \frac{\alpha_{i+1}}{\alpha_i} - 1 \right)\mathbf{x}_i + \left( \sigma_{i+1} - \frac{\alpha_{i+1}}{\alpha_i}\sigma_i \right)\varepsilon_\theta(x_i, i) \right] \\ & + \left[ \left( \frac{\alpha_{i-1}}{\alpha_i} - 1 \right)\mathbf{x}_i + \left( \sigma_{i-1} - \frac{\alpha_{i-1}}{\alpha_i}\sigma_i \right)\varepsilon_\theta(\mathbf{x}_i, i) \right] \\ =& \gamma\mathbf{x}_{i+1} + \left( \frac{\alpha_{i-1}}{\alpha_i} - \gamma\frac{\alpha_{i+1}}{\alpha_i} \right)\mathbf{x}_i + \left[ \sigma_{i-1} - \frac{\alpha_{i-1}}{\alpha_i}\sigma_i - \gamma\left( \sigma_{i+1} - \frac{\alpha_{i+1}}{\alpha_i}\sigma_i \right) \right]\varepsilon_\theta(\mathbf{x}_i, i). \end{aligned} \tag{24}$$

By rearranging the BDIA (24), the inversion of BDIA is

$$\begin{aligned} \mathbf{x}_{i+1} =& \mathbf{x}_{i-1}/\gamma + (1 - 1/\gamma)\mathbf{x}_i + \Delta\left(i \to i+1|\mathbf{x}_t\right) - (1/\gamma)\Delta\left(i \to i-1|\mathbf{x}_i\right), \\ =& \frac{1}{\gamma}\mathbf{x}_{i-1} + \left( 1 - \frac{1}{\gamma} \right)\mathbf{x}_i + \left[ \left( \frac{\alpha_{i+1}}{\alpha_i} - 1 \right)\mathbf{x}_i + \left( \sigma_{i+1} - \frac{\alpha_{i+1}}{\alpha_i}\sigma_i \right)\varepsilon_\theta(x_i, i) \right] \\ & - \frac{1}{\gamma}\left[ \left( \frac{\alpha_{i-1}}{\alpha_i} - 1 \right)\mathbf{x}_i + \left( \sigma_{i-1} - \frac{\alpha_{i-1}}{\alpha_i}\sigma_i \right)\varepsilon_\theta(\mathbf{x}_i, i) \right] \\ =& \frac{1}{\gamma}\mathbf{x}_{i-1} + \left( \frac{\alpha_{i+1}}{\alpha_i} - \frac{1}{\gamma}\frac{\alpha_{i-1}}{\alpha_i} \right)\mathbf{x}_i + \left[ \left( \sigma_{i+1} - \frac{\alpha_{i+1}}{\alpha_i}\sigma_i \right) - \frac{1}{\gamma}\left( \sigma_{i-1} - \frac{\alpha_{i-1}}{\alpha_i}\sigma_i \right) \right]\varepsilon_\theta(\mathbf{x}_i, i). \end{aligned} \tag{25}$$

## A.3    Continuity Assumption and Other Mathematical Remarks

**Continuity Assumption**    Much of our Local Truncation Error (LTE) analysis such as Proposition 1 and 4, is built on the Taylor expansion, which requires that the noise predictor satisfies the necessary continuity conditions. Therefore, we establish the following continuity assumption:

**Assumption 2.** *Denote* $\mathcal{E}_\theta(\bar{\sigma}_t) = \bar{\varepsilon}_\theta(\bar{\mathbf{x}}(t), \bar{\sigma}_t)$, *assume* $\mathcal{E}_\theta(\bar{\sigma}_t)$ *is continuous w.r.t.* $\bar{\sigma}_t$ :

$$\mathcal{E}_\theta(\bar{\sigma}_t) \in C^\infty(\mathbb{R}, \mathbb{R}^n). \tag{26}$$

This assumption can be met by selecting a differentiable activation design in the noise predictor U-Net [49].

**Variable of IVP** Here, we further wish to clarify that the notation $\bar{\varepsilon}_\theta(\bar{\mathbf{x}}(t), \bar{\sigma}_t) \equiv \varepsilon_\theta(\mathbf{x}(t), t)$ presented in Section 3 is well-defined. This is because there exists a bijective relationship between $\bar{\sigma}_t$ and $t$, and $\bar{\mathbf{x}}(t)$ is simply a scaled version of $\mathbf{x}(t)$.

**Singularity Issue** In Assumption 1, we do not consider the singularity points at $t = 0$ and $t = 1$ because these points can lead to unusual performance of the noise predictor as discussed in [74]. In fact, our numerical method is minimally affected by these singularity points, thus making Assumption 1 reasonable.

### A.4 Detailed Formulation of 3-step BELM

For 3-step BELM, we got five coefficients in the formulation:

$$\bar{\mathbf{x}}_{i-1} = a_{i,3}\bar{\mathbf{x}}_{i+2} + a_{i,2}\bar{\mathbf{x}}_{i+1} + a_{i,1}\bar{\mathbf{x}}_i + b_{i,2}h_{i+1}\bar{\varepsilon}_\theta(\bar{\mathbf{x}}_{i+1}, \bar{\sigma}_{i+1}) + b_{i,1}h_i\bar{\varepsilon}_\theta(\bar{\mathbf{x}}_i, \bar{\sigma}_i). \tag{27}$$

Follow the idea of Proposition 4, The local truncation error of the 3-step BELM diffusion sampler (27) $\tau_i$ can be accurate up to the fifth order of step sizes $\tau_i = \mathcal{O}\left((h_i + h_{i+1} + h_{i+2})^5\right)$ by setting coefficients as the following linear system

$$\begin{bmatrix} 1 & 1 & 1 & 0 & 0 \\ h_i & h_i + h_{i+1} & h_i + h_{i+1} + h_{i+2} & h_i & h_{i+1} \\ \frac{1}{2}h_i^2 & \frac{1}{2}(h_i + h_{i+1})^2 & \frac{1}{2}(h_i + h_{i+1} + h_{i+2})^2 & h_i^2 & h_{i+1}(h_i + h_{i+1}) \\ \frac{1}{6}h_i^3 & \frac{1}{6}(h_i + h_{i+1})^3 & \frac{1}{6}(h_i + h_{i+1} + h_{i+2})^3 & \frac{1}{2}h_i^3 & \frac{1}{2}h_{i+1}(h_i + h_{i+1})^2 \\ \frac{1}{24}h_i^4 & \frac{1}{24}(h_i + h_{i+1})^4 & \frac{1}{24}(h_i + h_{i+1} + h_{i+2})^4 & \frac{1}{6}h_i^4 & \frac{1}{6}h_{i+1}(h_i + h_{i+1})^3 \end{bmatrix} \begin{bmatrix} a_{i,1} \\ a_{i,2} \\ a_{i,3} \\ b_{i,1} \\ b_{i,2} \end{bmatrix} = \begin{bmatrix} 1 \\ 0 \\ 0 \\ 0 \\ 0 \end{bmatrix}. \tag{28}$$

There is no linear dependence between any two equations in (28). Through a calculation by hands or equation-solving tools like Matlab [27], the linear system above yields the unique solution provided below, which can be verified by readers.

$$\begin{bmatrix} a_{i,1} \\ a_{i,2} \\ a_{i,3} \\ b_{i,1} \\ b_{i,2} \end{bmatrix} = \begin{bmatrix} \frac{-((h_i+h_{i+1})^2(3h_i^2h_{i+1}+2h_i^2h_{i+2}+2h_ih_{i+1}^2+4h_ih_{i+1}h_{i+2}+2h_ih_{i+2}^2-h_{i+1}^3-2h_{i+1}^2h_{i+2}-h_{i+1}h_{i+2}^2))}{h_{i+1}^3(h_{i+1}+h_{i+2})^2} \\ \frac{(h_i^2(-h_i^2h_{i+1}+2h_i^2h_{i+2}-2h_ih_{i+1}^2+4h_ih_{i+1}h_{i+2}+2h_ih_{i+2}^2-h_{i+1}^3+2h_{i+1}^2h_{i+2}+3h_{i+1}h_{i+2}^2))}{h_{i+1}^3h_{i+2}^2} \\ \frac{h_i^2(h_i+h_{i+1})^2}{h_{i+2}^2(h_{i+1}+h_{i+2})^2} \\ \frac{-((h_i+h_{i+1})^2(h_i+h_{i+1}+h_{i+2}))}{h_{i+1}^2(h_{i+1}+h_{i+2})} \\ \frac{-(h_i^2(h_i+h_{i+1})(h_i+h_{i+1}+h_{i+2}))}{h_{i+1}^3h_{i+2}} \end{bmatrix}. \tag{29}$$

### A.5 Detailed Formulation of k-step BELM

For general k-step BELM, we got $2k - 1$ coefficients in the formulation:

$$\bar{\mathbf{x}}_{i-1} = \sum_{j=1}^{k} a_{i,j} \cdot \bar{\mathbf{x}}_{i-1+j} + \sum_{j=1}^{k-1} b_{i,j} \cdot h_{i-1+j} \cdot \bar{\varepsilon}_\theta(\bar{\mathbf{x}}_{i-1+j}, \bar{\sigma}_{i-1+j}). \tag{30}$$

Following the derivation of 2-step case, we first applying the Taylor's expansion to $\bar{\mathbf{x}}_{i-1+j}$ and $\bar{\varepsilon}_\theta(\bar{\mathbf{x}}_{i-1+j}, \bar{\sigma}_{i-1+j})$:

$$\sum_{j=1}^{k} a_{i,j} \cdot \bar{\mathbf{x}}_{i-1+j} + \sum_{j=1}^{k-1} b_{i,j} \cdot h_{i-1+j} \cdot \bar{\varepsilon}_\theta(\bar{\mathbf{x}}_{i-1+j}, \bar{\sigma}_{i-1+j})$$

$$= \sum_{j=1}^{k} a_{i,j} \left( \bar{\mathbf{x}}_{i-1} + \sum_{l=1}^{2k-2} \frac{1}{(l)!} \left( \sum_{m=0}^{j-1} h_{i+m} \right)^l \bar{\varepsilon}_\theta^{(l-1)} (\bar{\mathbf{x}}(t_{i-1}), \bar{\sigma}_{i-1}) \right)$$

$$+ \sum_{j=1}^{k-1} b_{i,j} h_{i-1+j} \left( \sum_{l=1}^{2k-2} \frac{1}{(l-1)!} \left( \sum_{m=0}^{j-1} h_{i+m} \right)^{l-1} \bar{\varepsilon}_\theta^{(l-1)} (\bar{\mathbf{x}}(t_{i-1}), \bar{\sigma}_{i-1}) \right)$$

$$+ \mathcal{O}\left( \left( \sum_{m=0}^{k-1} h_{i+m} \right)^{(2k-1)} \right) \tag{31}$$

$$= \sum_{j=1}^{k} a_{i,j} \bar{\mathbf{x}}_{i-1} + \sum_{l=1}^{2k-2} \left( \frac{1}{(l!)} \sum_{j=1}^{k} a_{i,j} \left( \sum_{m=1}^{j} h_{i+m-1} \right)^l \right) \bar{\varepsilon}_\theta^{(l-1)} (\bar{\mathbf{x}}(t_{i-1}), \bar{\sigma}_{i-1})$$

$$+ \sum_{l=1}^{2k-2} \left( \frac{1}{((l-1)!)} \sum_{j=1}^{k-1} b_{i,j} h_{i+j-1} \left( \sum_{m=1}^{j} h_{i+m-1} \right)^{l-1} \right) \bar{\varepsilon}_\theta^{(l-1)} (\bar{\mathbf{x}}(t_{i-1}), \bar{\sigma}_{i-1})$$

$$+ \mathcal{O}\left( \left( \sum_{m=0}^{k-1} h_{i+m} \right)^{(2k-1)} \right).$$

Thus, the optimal coefficient can be computed by:

$$\mathbf{A}_{(2k-1)\times(2k-1)} \begin{bmatrix} a_{i,1} \\ \vdots \\ a_{i,k} \\ b_{i,1} \\ \vdots \\ b_{i,k-1} \end{bmatrix}_{2k-1} = \begin{bmatrix} 1 \\ 0 \\ \vdots \\ 0 \end{bmatrix}_{2k-1}. \tag{32}$$

where $\mathbf{A} = [\mathbf{A}_1 \mid \mathbf{A}_2]$, and

$$\mathbf{A}_1 = \begin{pmatrix} 1 & 1 \cdots\cdots\cdots\cdots\cdots\cdots 1 \\ h_i & h_i + h_{i+1} \cdots\cdots\cdots\cdots \sum_{j=0}^{k-1} h_{i+j} \\ \frac{1}{2}h_i^2 \cdots\cdots\cdots & \frac{1}{2}(h_i + h_{i+1})^2 \cdots\cdots\cdots \frac{1}{2}\left(\sum_{j=0}^{k-1} h_{i+j}\right)^2 \\ \vdots \qquad\qquad\ddots\cdots\cdots & \qquad\qquad\ddots\cdots\cdots \qquad\qquad \vdots \\ \frac{1}{(2k-2)!}h_i^{2k-2} & \frac{1}{(2k-2)!}(h_i + h_{i+1})^{2k-2} \cdots \frac{1}{(2k-2)!}\left(\sum_{j=0}^{k-1} h_{i+j}\right)^{2k-2} \end{pmatrix}, \tag{33}$$

$$\mathbf{A}_2 = \begin{pmatrix} 0 & 0 \cdots\cdots\cdots\cdots\cdots\cdots\cdots 0 \\ h_i & h_{i+1} \cdots\cdots\cdots\cdots\cdots\cdots h_{i+k-2} \\ h_i^2 & h_{i+1}(h_i + h_{i+1}) \cdots\cdots\cdots h_{i+k-2}\left(\sum_{j=0}^{k-2} h_{i+j}\right) \\ \frac{1}{2}h_i^3 \cdots\cdots & \frac{1}{2}h_{i+1}(h_i + h_{i+1})^2 \cdots\cdots \frac{1}{2}h_{i+k-2}\left(\sum_{j=0}^{k-2} h_{i+j}\right)^2 \\ \vdots \qquad\ddots\cdots\cdots & \qquad\qquad\ddots\cdots\cdots \qquad\qquad \vdots \\ \frac{1}{(2k-3)!}h_i h_i^{2k-3} & \frac{1}{(2k-3)!}h_i(h_i + h_{i+1})^{2k-3} \cdots \frac{1}{(2k-3)!}h_{i+k-2}\left(\sum_{j=0}^{k-2} h_{i+j}\right)^{2k-3} \end{pmatrix}. \tag{34}$$

## A.6 Definitions of Consistency

**Consistency** The consistency property refers to the ability of the method to accurately represent the IVP equation it's trying to solve. More specifically, a method is said to be consistent if, as the step size approaches zero, the difference between the numerical method and the exact differential equation also approaches zero.

> **Definition 4.** *The LM method* (12) *is **consistent** if for every function* $y \in C^1[t_0, t_0 + T]$
> $$\lim_{h \to 0} \sum_{i=k}^{N-1} \|\tau_i\| = 0. \tag{35}$$

## A.7 BDIA as a Sub-Optimal Special Case of BELM

the updating rule of BDIA write:

$$\mathbf{x}_{i-1} = \gamma \mathbf{x}_{i+1} + \left( \frac{\alpha_{i-1}}{\alpha_i} - \gamma \frac{\alpha_{i+1}}{\alpha_i} \right) \mathbf{x}_i + \left[ \sigma_{i-1} - \frac{\alpha_{i-1}}{\alpha_i} \sigma_i - \gamma \left( \sigma_{i+1} - \frac{\alpha_{i+1}}{\alpha_i} \sigma_i \right) \right] \boldsymbol{\varepsilon}_\theta(\mathbf{x}_i, i). \tag{36}$$

With the same alpha, scaled sigma and stepsize schedule as the BELM, the BDIA update (36) have an equivalent **bidirectional explicit** linear multi-step form with an easy rearrangement,

$$\bar{\mathbf{x}}_{i-1} = a_{i,2} \cdot \bar{\mathbf{x}}_{i+1} + a_{i,1} \cdot \bar{\mathbf{x}}_i + b_{i,1} \cdot h_i \cdot \bar{\boldsymbol{\varepsilon}}_\theta(\bar{\mathbf{x}}_i, \bar{\sigma}_i), \tag{37}$$

where

$$a_{i,2} = \gamma \frac{\alpha_{i+1}}{\alpha_{i-1}}, \qquad a_{i,1} = 1 - \gamma \frac{\alpha_{i+1}}{\alpha_{i-1}}, \qquad b_{i,1} = -1 - \gamma \frac{\alpha_{i+1}}{\alpha_{i-1}} \frac{h_{i+1}}{h_i}. \tag{38}$$

Thus we find that BDIA is indeed a special case of our BELM framework.

## A.8 EDICT as a Sub-Optimal Special Case of BELM

In this section, we will demonstrate that a sequence of $\{\mathbf{x}_i\}$, $\{\mathbf{y}_i\}$, $\{\mathbf{y}_i^{inter}\}$, and $\{\mathbf{x}_i^{inter}\}$, where $i \in [N \dots 1]$, generated by EDICT (9), indeed corresponds to a sequence of $\mathbf{z}_j$, where $j \in [4N \dots 1]$, produced by a special BELM. The EDICT updates as follows:

$$\begin{cases} \mathbf{x}_i^{inter} = \dfrac{\alpha_{i-1}}{\alpha_i} \mathbf{x}_i + \left( \sigma_{i-1} - \dfrac{\alpha_{i-1}}{\alpha_i} \sigma_i \right) \boldsymbol{\varepsilon}_\theta(\mathbf{y}_i, i), \\[2mm] \mathbf{y}_i^{inter} = \dfrac{\alpha_{i-1}}{\alpha_i} \mathbf{y}_i + \left( \sigma_{i-1} - \dfrac{\alpha_{i-1}}{\alpha_i} \sigma_i \right) \boldsymbol{\varepsilon}_\theta(\mathbf{x}(t)^{inter}, i), \\[2mm] \mathbf{x}_{i-1} = p \mathbf{x}_i^{inter} + (1-p) \mathbf{y}_i^{inter}, \\[2mm] \mathbf{y}_{i-1} = p \mathbf{y}_i^{inter} + (1-p) \mathbf{x}_{i-1}. \end{cases} \tag{39}$$

transfer $\mathbf{x}_i$ to $\mathbf{z}_{4l}$, $\mathbf{y}_i$ to $\mathbf{z}_{4l-1}$, $\mathbf{x}_i^{inter}$ to $\mathbf{z}_{4l-2}$ and $\mathbf{y}_i^{inter}$ to $\mathbf{z}_{4l-3}$,

$$\begin{cases} \mathbf{z}_{4l-2} = \dfrac{\alpha_{i-1}}{\alpha_i} \mathbf{z}_{4l} + \left( \sigma_{i-1} - \dfrac{\alpha_{i-1}}{\alpha_i} \sigma_i \right) \boldsymbol{\varepsilon}_\theta(\mathbf{z}_{4l-1}, i), \\[2mm] \mathbf{z}_{4l-3} = \dfrac{\alpha_{i-1}}{\alpha_i} \mathbf{z}_{4l-1} + \left( \sigma_{i-1} - \dfrac{\alpha_{i-1}}{\alpha_i} \sigma_i \right) \boldsymbol{\varepsilon}_\theta(\mathbf{z}_{4l-2}, i), \\[2mm] \mathbf{z}_{4l-4} = p \mathbf{z}_{4l-2} + (1-p) \mathbf{z}_{4l-3}, \\[2mm] \mathbf{z}_{4l-5} = p \mathbf{z}_{4l-3} + (1-p) \mathbf{z}_{4l-4}. \end{cases} \tag{40}$$

We set alpha schedule to be

$$\alpha_{4l} = \alpha_i, \quad \alpha_{4l-1} = \alpha_i, \quad \alpha_{4l-2} = \sqrt{\alpha_i \alpha_{i-1}}, \quad \alpha_{4l-3} = \alpha_{i-1}. \tag{41}$$

Then we set sigma schedule to be

$$\sigma_{4l} = \sigma_i, \quad \sigma_{4l-1} = \sigma_i, \quad \sigma_{4l-2} = \frac{1}{2} \left( \sigma_i \frac{\sqrt{\alpha_{i-1}}}{\sqrt{\alpha_i}} + \sigma_{i-1} \frac{\sqrt{\alpha_i}}{\sqrt{\alpha_{i-1}}} \right), \quad \sigma_{4l-3} = \sigma_{i-1}. \tag{42}$$

Thus the scaled sigma writes

$$\bar{\sigma}_{4l} = \bar{\sigma}_{4l-1} = \frac{\sigma_i}{\alpha_i}, \quad \bar{\sigma}_{4l-2} = \frac{\sigma_i}{\alpha_i} + \frac{\sigma_{i-1}}{\alpha_{i-1}}, \quad \bar{\sigma}_{4l-3} = \frac{\sigma_{i-1}}{\alpha_{i-1}}. \tag{43}$$

And the stepsize schedule will be

$$h_{4l} = 0, \quad h_{4l-1} = \frac{1}{2}\left(\frac{\sigma_i}{\alpha_i} - \frac{\sigma_{i-1}}{\alpha_{i-1}}\right), \quad h_{4l-2} = \frac{1}{2}\left(\frac{\sigma_i}{\alpha_i} - \frac{\sigma_{i-1}}{\alpha_{i-1}}\right), \quad h_{4l-3} = 0. \tag{44}$$

With easy substitution, the EDICT update (40) have an equivalent **bidirectional explicit** linear multi-step form:

$$\bar{\mathbf{z}}_{j-1} = a_{j,2} \cdot \bar{\mathbf{z}}_{j+1} + a_{j,1} \cdot \bar{\mathbf{z}}_j + b_{j,1} \cdot h_j \cdot \bar{\varepsilon}_\theta(\bar{\mathbf{z}}_j, \bar{\sigma}_j) \tag{45}$$

where the coefficients take the following piece-wise function form:

$$a_{j,2} = \begin{cases} p\dfrac{\sqrt{\alpha_{i+1}}}{\sqrt{\alpha_i}}, j = 4l \\ p, j = 4l-1 \\ \dfrac{\sqrt{\alpha_{i-1}}}{\sqrt{\alpha_i}}, j = 4l-2 \\ 1, j = 4l-3 \end{cases} \quad a_{j,1} = \begin{cases} 1-p, j = 4l \\ 1-p, j = 4l-1 \\ 0, j = 4l-2 \\ 0, j = 4l-3 \end{cases} \quad b_{j,1} = \begin{cases} 0, j = 4l \\ 0, j = 4l-1 \\ -2\dfrac{\sqrt{\alpha_{i-1}}}{\sqrt{\alpha_i}}, j = 4l-2 \\ -2, j = 4l-3 \end{cases} \tag{46}$$

Despite the formulation of (9) being subject to cyclic variations, the variable $a_{j,2}$ consistently remains non-zero, thereby satisfying the conditions of the BELM framework. Consequently, EDICT can indeed be considered a special case within our BELM framework.

## A.9 Order of Accuracy

In this section, we further explore the order of accuracy of DDIM, EDICT, BDIA, and our proposed O-BELM. Our findings indicate that O-BELM achieves the superior order of accuracy among these methods. Intuitively, the order of accuracy provides insight into which functional class of the IVP can be accurately approximated by a given method.

**Definition 5.** *The method (14) is said to have **order of accuracy p** if $p$ is the largest positive integer such that there exist constants $K$ and $h^*$ such that for all $i$,*

$$\|\tau_i\| \le K h^{p+1}, \quad for \quad 0 < h < h^*. \tag{47}$$

**Proposition 6.** *The BELM diffusion sampler (15) is with **second-order accuracy**; The DDIM diffusion sampler (7) is with **first-order accuracy**; The EDICT diffusion sampler (9) is with **zero-order accuracy**; The BDIA diffusion sampler (10) is with **first-order accuracy**.*

*Proof.* This proposition can be directly inferred from the Definition 5, in conjunction with Proposition 1 and 4. □

**Remark 2.** *Though the order of accuracy of BDIA is the same as DDIM to be 1, its step size in local error of BDIA is about twice that of DDIM. This theoretical result confirms the experimental observation that the sampling quality of BDIA sometimes is inferior to that of DDIM.*

## A.10 Further Theoretical Properties of DDIM

We have also conducted an analysis of global stability and convergence for DDIM. It is apparent that the success of DDIM fundamentally stems from its nice theoretical property. Our O-BELM preserves these excellent theoretical properties of DDIM and maintains high-quality sampling performance.

**Proposition 7.** *The DDIM diffusion sampler (7) is (a) **zero-stable** and (b) **globally convergent**.*

## A.11 Pseudocode for O-BELM Sampling Process

To more effectively elucidate the implementation of O-BELM, we provide the pseudocode for O-BELM in Algorithm 1. Upon examination, it is apparent that the implementation of O-BELM requires little modifications compared to DDIM, thus facilitating its easy portability to pretrained models.

---

**Algorithm 1** O-BELM sampling process

---

1: **Input**: pretrained noise predictor $\varepsilon_\theta$, number of timesteps $N$, noise schedule $\{\alpha_t\}$ and $\{\sigma_t\}$, x_list = [].
2: Sample $\mathbf{x}_N \sim \mathcal{N}(0, \sigma_{t_N} I)$.
3: x_list.append($\mathbf{x}_N$)
4: **for** $i = N, N - 1, ..., 1$ **do**
5:    **if** $i < N$ **then**
6:       Calculate $h_i$, $a_{i,1}$, $a_{i,2}$ and $b_{i,1}$ according to (53).
7:       $\mathbf{x}_{i-1} = a_{i,1}\text{x\_list[-1]} + a_{i,2}\text{x\_list[-2]} + b_{i,1}h_i\varepsilon_\theta(\text{x\_list[-1]}, i)$
8:    **else**
9:       $\mathbf{x}_{i-1} = \frac{\alpha_{i-1}}{\alpha_i}\text{x\_list[-1]} + \left(\sigma_{i-1} - \frac{\alpha_{i-1}}{\alpha_i}\sigma_i\right)\varepsilon_\theta(\text{x\_list[-1]}, i)$.
10:    **end if**
11:    x_list.append($\mathbf{x}_{i-1}$)
12: **end for**
13: **Output**: x_list

---

# B Proofs

## B.1 Proof of Proposition 2

*Proof.* We demonstrate the exact inversion property of BELM (14) by initially establishing that its local reconstruction error is zero. Assuming that we have already obtained $\{\bar{\mathbf{x}}_{i-1+j}\}_{j=1}^{k}$, we compute $\bar{\mathbf{x}}_{i-1}$ in accordance with (14), as follows:

$$\bar{\mathbf{x}}_{i-1} = \sum_{j=1}^{k} a_{i,j} \cdot \bar{\mathbf{x}}_{i-1+j} + \sum_{j=1}^{k-1} b_{i,j} \cdot h_{i-1+j} \cdot \bar{\varepsilon}_\theta(\bar{\mathbf{x}}_{i-1+j}, \bar{\sigma}_{i-1+j}), \qquad (48)$$

and we will use $\{\bar{\mathbf{x}}_{i-1+j}\}_{j=0}^{k-1}$ to reconstruct $\tilde{\bar{\mathbf{x}}}_{i-1+k}$ according to (13), as follows:

$$\tilde{\bar{\mathbf{x}}}_{i-1+k} = \frac{1}{a_{i,k}} \cdot \bar{\mathbf{x}}_{i-1} - \sum_{j=1}^{k-1} \frac{a_{i,j}}{a_{i,k}} \cdot \bar{\mathbf{x}}_{i-1+j} + \sum_{j=0}^{k} \frac{b_{i,j}}{a_{i,k}} \cdot h_{i-1+j} \cdot \bar{\varepsilon}_\theta(\bar{\mathbf{x}}_{i-1+j}, \bar{\sigma}_{i-1+j}). \qquad (49)$$

The local reconstruction error, defined as the difference between $\tilde{\bar{\mathbf{x}}}_{i-1+k}$ and $\bar{\mathbf{x}}_{i-1+k}$, can be calculated and is found to be zero. Furthermore, global exact inversion can be inferred from local exact inversion through the application of Mathematical Induction (MI). $\qquad\square$

## B.2 Proof of Proposition 3

*Proof.* The Local Truncation Error (LTE) of the BELM diffusion sampler (15) can be computed by substituting $\bar{\mathbf{x}}(t_i)$, $\bar{\mathbf{x}}(t_{i+1})$, and $\bar{\varepsilon}_\theta(\bar{\mathbf{x}}(t_i), \bar{\sigma}_i)$ in(16) with their corresponding Taylor expansions at

$\bar{\sigma}_{i-1}$ as follows:

$$
\begin{aligned}
\tau_i =& \bar{\mathbf{x}}(t_{i-1}) - a_{i,1} \cdot \bar{\mathbf{x}}(t_i) - a_{i,2} \cdot \bar{\mathbf{x}}(t_{i+1}) - b_{i,1} \cdot h_i \cdot \bar{\varepsilon}_\theta(\bar{\mathbf{x}}(t_i), \bar{\sigma}_i) \\
=& \bar{\mathbf{x}}(t_{i-1}) - a_{i,1} \left[ \bar{\mathbf{x}}(t_{i-1}) + \frac{\bar{\varepsilon}_\theta(\bar{\mathbf{x}}(t_{i-1}), \bar{\sigma}_{i-1})}{1!}(h_i) \right. \\
& \left. + \frac{\nabla_{\bar{\sigma}_{i-1}} \bar{\varepsilon}_\theta(\bar{\mathbf{x}}(t_{i-1}), \bar{\sigma}_{i-1})}{2!}(h_i)^2 + \mathcal{O}\left(h_i{}^3\right) \right] \\
& - a_{i,2} \left[ \bar{\mathbf{x}}(t_{i-1}) + \frac{\bar{\varepsilon}_\theta(\bar{\mathbf{x}}(t_{i-1}), \bar{\sigma}_{i-1})}{1!}(h_i + h_{i+1}) + \right. \\
& \left. \frac{\nabla_{\bar{\sigma}_{i-1}} \bar{\varepsilon}_\theta(\bar{\mathbf{x}}(t_{i-1}), \bar{\sigma}_{i-1})}{2!}(h_i + h_{i+1})^2 + \mathcal{O}\left((h_i + h_{i+1})^3\right) \right] \\
& - b_{i,1} \cdot h_i \left[ \bar{\varepsilon}_\theta(\bar{\mathbf{x}}(t_{i-1}), \bar{\sigma}_{i-1}) + \frac{\nabla_{\bar{\sigma}_{i-1}} \bar{\varepsilon}_\theta(\bar{\mathbf{x}}(t_{i-1}), \bar{\sigma}_{i-1})}{1!}(h_i) + \mathcal{O}\left(h_i{}^2\right) \right] \\
=& (1 - a_{i,1} - a_{i,2}) \bar{\mathbf{x}}(t_{i-1}) \\
& + [-a_{i,1}h_i - a_{i,2}(h_i + h_{i+1}) - b_{i,1} \cdot h_i] \cdot \bar{\varepsilon}_\theta(\bar{\mathbf{x}}(t_{i-1}), \bar{\sigma}_{i-1}) \\
& + \left[ -\frac{a_{i,1}}{2} \cdot h_i^2 - \frac{a_{i,2}}{2}(h_i + h_{i+1})^2 - b_{i,1} \cdot h_i^2 \right] \cdot \nabla_{\bar{\sigma}_{i-1}} \bar{\varepsilon}_\theta(\bar{\mathbf{x}}(t_{i-1}), \bar{\sigma}_{i-1}) \\
& + \mathcal{O}\left((h_i + h_{i+1})^3\right).
\end{aligned}
\tag{50}
$$

$\square$

## B.3 Proof of Proposition 4

*Proof.* In the (17) of Proposition 3, we have three degrees of freedom: $a_{i,1}$, $a_{i,2}$, and $b_{i,1}$ in the LTE of BELM (15). Therefore, the highest order that $\tau_i$ can achieve is three, under the condition that:

$$
\begin{cases}
1 - a_{i,1} - a_{i,2} = 0, \\
- a_{i,1}h_i - a_{i,2}(h_i + h_{i+1}) - b_{i,1} \cdot h_i = 0, \\
- \frac{a_{i,1}}{2} \cdot h_i^2 - \frac{a_{i,2}}{2}(h_i + h_{i+1})^2 - b_{i,1} \cdot h_i^2 = 0.
\end{cases}
\tag{51}
$$

whose matrix form writes

$$
\begin{bmatrix}
1 & 1 & 0 \\
h_i & (h_i + h_{i+1}) & h_i \\
\frac{1}{2}h_i^2 & \frac{1}{2}(h_i + h_{i+1})^2 & h_i^2
\end{bmatrix}
\begin{bmatrix}
a_{i,1} \\
a_{i,2} \\
b_{i,1}
\end{bmatrix}
=
\begin{bmatrix}
1 \\
0 \\
0
\end{bmatrix}.
\tag{52}
$$

There is no linear dependence between any two equations in (51). Through a straightforward calculation, the linear system above yields the unique solution provided below, which can be verified by readers.

$$
a_{i,1} = \frac{h_{i+1}^2 - h_i^2}{h_{i+1}^2}, \quad a_{i,2} = \frac{h_i^2}{h_{i+1}^2}, \quad b_{i,1} = -\frac{h_i + h_{i+1}}{h_{i+1}}.
\tag{53}
$$

$\square$

## B.4 Proof of Corollary 1

### B.4.1 LTE of DDIM

**Proposition 8.** *The LTE* $\mathbf{e}_i$ *of DDIM sampler* (7) *is* $\mathcal{O}\left(\alpha_{i-1}h_i{}^2\right)$.

*Proof.* By applying the Taylor expansion and substitute into the DDIM formulation (7), we can calculate the local error of DDIM on $\mathbf{x}_i$ as following.

$$
\begin{aligned}
\mathbf{e}_i =& \mathbf{x}(t_{i-1}) - \frac{\alpha_{i-1}}{\alpha_i}\mathbf{x}(t_i) - \left(\sigma_{i-1} - \frac{\alpha_{i-1}}{\alpha_i}\sigma_i\right)\bar{\varepsilon}_\theta\left(\bar{\mathbf{x}}(t_i), \bar{\sigma}_i\right) \\
=& \mathbf{x}(t_{i-1}) - \frac{\alpha_{i-1}}{\alpha_i}\alpha_i\left(\frac{\mathbf{x}(t_{i-1})}{\alpha_{i-1}} + \frac{\bar{\varepsilon}_\theta\left(\bar{\mathbf{x}}(t_{i-1}), \bar{\sigma}_{i-1}\right)}{1!}(h_i) + \mathcal{O}\left(h_i{}^2\right)\right) \\
& - (-h_i)\alpha_{i-1}\left(\bar{\varepsilon}_\theta\left(\bar{\mathbf{x}}(t_{i-1}), \bar{\sigma}_{i-1}\right) + \mathcal{O}\left(h_i\right)\right) \\
=& \mathcal{O}\left(\alpha_{i-1}h_i{}^2\right).
\end{aligned}
\tag{54}
$$

$\square$

### B.4.2 LTE of BDIA

**Proposition 9.** *The LTE $\mathbf{e}_i$ of BDIA sampler* (10) *is $\mathcal{O}\left(\alpha_{i-1}(h_i + h_{i+1})^2\right)$ for any fixed $\gamma \in [0, 1]$.*

*Proof.* By applying the Taylor expansion and substitute into the BDIA formulation (10), we can calculate the local error of BDIA on $\mathbf{x}_i$ as following.

$$
\begin{aligned}
\mathbf{e}_i =& \mathbf{x}(t_{i-1}) - \gamma\mathbf{x}(t_{i+1}) - \left(\frac{\alpha_{i-1}}{\alpha_i} - \gamma\frac{\alpha_{i+1}}{\alpha_i}\right)\mathbf{x}(t_i) \\
& - \left[\sigma_{i-1} - \frac{\alpha_{i-1}}{\alpha_i}\sigma_i - \gamma\left(\sigma_{i+1} - \frac{\alpha_{i+1}}{\alpha_i}\sigma_i\right)\right]\bar{\varepsilon}_\theta\left(\bar{\mathbf{x}}(t_i), \bar{\sigma}_i\right) \\
=& \mathbf{x}(t_{i-1}) - \gamma\alpha_{i+1}\left[\frac{\mathbf{x}(t_{i-1})}{\alpha_{i-1}} + \frac{\bar{\varepsilon}_\theta\left(\bar{\mathbf{x}}(t_{i-1}), \bar{\sigma}_{i-1}\right)}{1!}(h_i + h_{i+1}) + \right. \\
& \left. \frac{\nabla_{\bar{\sigma}_{i-1}}\bar{\varepsilon}_\theta\left(\bar{\mathbf{x}}(t_{i-1}), \bar{\sigma}_{i-1}\right)}{2!}(h_i + h_{i+1})^2 + \mathcal{O}\left((h_i + h_{i+1})^3\right)\right] \\
& - \left(\frac{\alpha_{i-1}}{\alpha_i} - \gamma\frac{\alpha_{i+1}}{\alpha_i}\right)\alpha_i\left[\frac{\mathbf{x}(t_{i-1})}{\alpha_{i-1}} + \frac{\bar{\varepsilon}_\theta\left(\bar{\mathbf{x}}(t_{i-1}), \bar{\sigma}_{i-1}\right)}{1!}(h_i)\right. \\
& \left. + \frac{\nabla_{\bar{\sigma}_{i-1}}\bar{\varepsilon}_\theta\left(\bar{\mathbf{x}}(t_{i-1}), \bar{\sigma}_{i-1}\right)}{2!}(h_i)^2 + \mathcal{O}\left(h_i{}^3\right)\right] \\
& - \left[\sigma_{i-1} - \frac{\alpha_{i-1}}{\alpha_i}\sigma_i - \gamma\left(\sigma_{i+1} - \frac{\alpha_{i+1}}{\alpha_i}\sigma_i\right)\right]\left[\bar{\varepsilon}_\theta\left(\bar{\mathbf{x}}(t_{i-1}), \bar{\sigma}_{i-1}\right)\right. \\
& \left. + \frac{\nabla_{\bar{\sigma}_{i-1}}\bar{\varepsilon}_\theta\left(\bar{\mathbf{x}}(t_{i-1}), \bar{\sigma}_{i-1}\right)}{1!}(h_i) + \mathcal{O}\left(h_i{}^2\right)\right] \\
=& \left(-\frac{\gamma}{2}\alpha_{i+1}h_{i+1}^2 + \frac{3}{2}\alpha_{i-1}h_i^2\right)\nabla_{\bar{\sigma}_{i-1}}\bar{\varepsilon}_\theta\left(\bar{\mathbf{x}}(t_{i-1}), \bar{\sigma}_{i-1}\right) + \mathcal{O}\left(h_i{}^3\right).
\end{aligned}
\tag{55}
$$

For a fixed $\gamma$, the term $\bar{\varepsilon}_\theta\left(\bar{\mathbf{x}}(t_{i-1}), \bar{\sigma}_{i-1}\right)$ cannot be eliminated for every $i$. This is due to the fact that the second-order term $-\frac{\gamma}{2}\alpha_{i+1}h_{i+1}^2 + \frac{3}{2}\alpha_{i-1}h_i^2$ is dynamic with respect to $i$. Consequently, the second-order local error will persist in the BDIA. $\square$

### B.4.3 LTE of EDICT

**Proposition 10.** *The LTE $\mathbf{e}_i$ of EDICT sampler* (9) *is $\mathcal{O}\left(\sqrt{\alpha_{i-1}}h_i\right)$ for any constant $p \in (0, 1)$.*

To prove the Proposition 10, we need first establish an order estimate lemma:

**Lemma 1.** *The term $\sqrt{\alpha_i} - \sqrt{\alpha_{i-1}}$ have order $\mathcal{O}\left(h_i\right)$*

*Proof.* Recall that we define $h_i$ to be $\bar{\sigma}_i - \bar{\sigma}_{i-1}$. In order to figure out the relation of $\sqrt{\alpha_i} - \sqrt{\alpha_{i-1}}$ w.r.t. $\bar{\sigma}_i - \bar{\sigma}_{i-1}$, we first use $\bar{\sigma}$ to represent $\sqrt{\alpha}$:

$$\bar{\sigma} = \frac{\sqrt{1-\alpha^2}}{\alpha}$$
$$\bar{\sigma}^2 = \frac{1-\alpha^2}{\alpha^2}$$
$$(\bar{\sigma}^2 + 1)\alpha^2 = 1 \tag{56}$$
$$\alpha = \sqrt{\frac{1}{\bar{\sigma}^2 + 1}}$$
$$\sqrt{\alpha} = (\bar{\sigma}^2 + 1)^{-\frac{1}{4}}.$$

We then discover that $\mathrm{d}\sqrt{\alpha} = -\frac{1}{2}(\bar{\sigma}^2 + 1)^{-\frac{5}{4}}\bar{\sigma}\mathrm{d}\bar{\sigma} \sim C\mathrm{d}\bar{\sigma}$. This implies that $\sqrt{\alpha_i} - \sqrt{\alpha_{i-1}}$ and $\bar{\sigma}_i - \bar{\sigma}_{i-1}$ are of the same order, which is $h_i$. $\qquad\square$

Now we can start to prove Proposition 10:

*Proof.* Since larger errors can absorb smaller ones, the $4l$ and $4l-2$ terms in (46) introduce errors in the zeroth order of the Taylor expansion. This is where the main error occurs. Both of these updates introduce an error of $\frac{\sqrt{\alpha_i} - \sqrt{\alpha_{i-1}}}{\sqrt{\alpha_i}}$ on $\bar{\mathbf{x}}_i$, which means that the error on $\mathbf{x}_i$ is $\sqrt{\alpha_i}\left(\sqrt{\alpha_i} - \sqrt{\alpha_{i-1}}\right)$. Therefore, according to Lemma 1, the error $e_i$ is of the order $\mathcal{O}\left(\sqrt{\alpha_i}h_i\right)$. $\qquad\square$

> **Remark 3.** *Please note that we have only established an error bound for EDICT based on the perspective of the linear multiplication method. There may be a tighter bound of EDICT on constants when viewed from the perspective of an interactive mixing system.*

## B.5   Proof of Proposition 5(a) and Proposition 7(a)

> **Assumption 3.** $\bar{\sigma}_i$ *is strictly concave w.r.t.* $i$.

$\bar{\sigma}_i$ w.r.t $i$ is a composition of $\bar{\sigma}_i$ w.r.t $\alpha_i$ and $\alpha_i$ w.r.t. $i$. $\bar{\sigma}_i = \frac{\sqrt{1-\alpha_i^2}}{\alpha_i}$ which is non-increasing and strictly convex. Thus Assumption 3 can be achieved by choosing schedule of $\alpha_i$ to be strictly convex w.r.t. $i$.

> **Lemma 2.** *There exist a real constant $C$ which is independent of $\alpha_t$ and $\sigma_t$, such that for every $i$, we have $|a_{i,1}| \leq C$, $|a_{i,2}| \leq C$ and $|b_{i,1}| \leq C$ in (15).*

We will use an variable-stepsize-variable-formula analogy of the root condition of Dahlquist [9] to prove the zero-stability of (15).

> **Theorem 1.** *[8, (3.10)] Define the root matrix of a LM 12 at step $i$ to be $\mathbf{R}_i$,*
>
> $$\mathbf{R}_i = \begin{pmatrix} a_{i,1} & \cdots\cdots\cdots & a_{i,k} \\ 1 & 0 & 0 \\ \vdots & \ddots\ddots\ddots & \vdots \\ 0 & 1 & 0 \end{pmatrix}. \tag{57}$$
>
> *If all coefficients can be bounded and there exists a regular matrix $\mathbf{H}$ such that for all $i$*
>
> $$\left\|\mathbf{H}^{-1}\mathbf{R}_i\mathbf{H}\right\|_1 \leq 1, \tag{58}$$
>
> *then the LM 12 is zero-stable.*

Finally, we start to give the proof of Proposition 5(a) under the Assumption 3.

**Proposition 11.** *The O-BELM diffusion sampler* (18) *is zero-stable.*

*Proof.* the root matrix of (15) writs

$$\mathbf{R}_i = \begin{bmatrix} \frac{h_{i+1}^2 - h_i^2}{h_{i+1}^2} & \frac{h_i^2}{h_{i+1}^2} \\ 1 & 0 \end{bmatrix}. \tag{59}$$

The Assumption 3 can reach to $\bar{\sigma}_{i+1} + \bar{\sigma}_{i-1} < 2\bar{\sigma}_i$, thus $h_{i+1} < h_i < 0$. Then we denote $\eta = \max_i \frac{h_i^2}{h_{i+1}^2} < 1$, by setting $\mathbf{H}$ as following

$$\mathbf{H} = \begin{bmatrix} 1 & \frac{2}{1-\eta} \\ 0 & \frac{2}{1-\eta} \end{bmatrix}, \tag{60}$$

then we can calculate that

$$\left\| \mathbf{H}^{-1}\mathbf{R}_i\mathbf{H} \right\|_1 = \left\| \begin{bmatrix} 1 & \frac{2}{1-\eta} \\ 0 & \frac{2}{1-\eta} \end{bmatrix}^{-1} \begin{bmatrix} \frac{h_{i+1}^2 - h_i^2}{h_{i+1}^2} & \frac{h_i^2}{h_{i+1}^2} \\ 1 & 0 \end{bmatrix} \begin{bmatrix} 1 & \frac{2}{1-\eta} \\ 0 & \frac{2}{1-\eta} \end{bmatrix} \right\|_1$$
$$= \max \left( \frac{|\frac{\eta}{2} - \frac{1}{2}||h_{i+1}|^2 + |h_i|^2}{|h_{i+1}|^2}, \frac{2|\frac{\eta}{2} - \frac{1}{2}|}{|\eta - 1|} \right), \tag{61}$$

where we can compute that

$$\frac{|\frac{\eta}{2} - \frac{1}{2}||h_{i+1}|^2 + |h_i|^2}{|h_{i+1}|^2}$$
$$= \left| \frac{\eta}{2} - \frac{1}{2} \right| + \frac{h_i^2}{h_{i+1}^2}$$
$$= \frac{1}{2} - \frac{\eta}{2} + \frac{h_i^2}{h_{i+1}^2}$$
$$< \frac{1}{2} - \frac{1}{2}\frac{h_i^2}{h_{i+1}^2} + \frac{h_i^2}{h_{i+1}^2}$$
$$= \frac{1}{2}\frac{h_i^2 + h_{i+1}^2}{h_{i+1}^2}$$
$$< 1, \tag{62}$$

and obviously

$$\frac{2|\frac{\eta}{2} - \frac{1}{2}|}{|\eta - 1|} = 1. \tag{63}$$

Consequently, we have the conclusion that for all $i$, the requirement of $\left\| \mathbf{H}^{-1}\mathbf{R}_i\mathbf{H} \right\|_1 \leq 1$ is satisfied. Thus due to Theorem 1, The BELM diffusion sampler (15) is zero-stable. $\square$

**Remark 4.** *Here, we present a very strong proof of Proposition 11 under Assumption 3, demonstrating that the iterative mapping of BELM constitutes a contraction mapping at each step $i$. However, it is important to note that in practical applications, even if Assumption 3 is not met at some step $i$, resulting in $\mathbf{R}_i$ not being contractive sometimes, global stability may still be achieved.*

The proof for Proposition 7(a) writes:

*Proof.* As DDIM can be seen as an explicit Euler method to the diffusion IVP, following the same reasoning of B.5, the root matrix of DDIM is $\mathbf{R}_i = \mathbf{I}$. Obviously, DDIM is zero-stable. $\square$

### B.6 Proof of Proposition 5(b) and Proposition 7(b)

To analyse the global convergence property of a method, we first need to analyse the consistency property of a method. Please look up the definition of consistency in Appendix A.6. We first establish the consistency of DDIM and BELM by the following theorem.

> **Theorem 2.** *[8, (2.5.1)] If a method have an order of accuracy 1 and all its coefficients is bounded by constant, then it is consistent.*

> **Lemma 3.** *The BELM diffusion sampler* (15) *is consistent.*

*Proof.* This lemma is a direct result of Lemma 2, Proposition 6 and Theorem 2. ☐

> **Lemma 4.** *The DDIM diffusion sampler* (7) *is consistent.*

*Proof.* In common choice of noise schedule, $\frac{\alpha_{i-1}}{\alpha_i}$ and $\sigma_{i-1} - \frac{\alpha_{i-1}}{\alpha_i}\sigma_i$ is bounded. Thus this lemma is a direct result of Theorem 2. ☐

After we establish the consistency of DDIM and BELM, we can prove their global convergence by a famous sufficiency of conditions for convergence.

> **Theorem 3.** *[3, p.342 (Theorem 406D)] A linear multistep method is convergent if it is consistent and zero-stable.*

With the help of Theorem 3, we can reach to Proposition 7(b) by Lemma 4 and Proposition 7(a); and reach to Proposition 5(b) by Lemma 3 and Proposition 11.

## C   Experiments Details and Extra Results

In these image tasks, we only apply our 2-step O-BELM, as it has been demonstrated that higher-order numerical methods can lead to strong oscillations in stiff spaces such as images [58, p.343]. However, the application of higher-order O-BELM in other domains of Diffusion Models (DMs) continues to hold promise.

For the sake of open accessibility, the dataset used in this paper is publicly available on the internet. We have included codes, accompanied by corresponding instructions, in the supplementary materials and plan to make them accessible on GitHub. However, our Stable Diffusion-related code is intricately interwoven with our proprietary business code, and we are in the process of decoupling the codebase. As soon as this task is completed, we will make the codes available on GitHub.

### C.1   Image Reconstruction

Figure 4 presents the reconstruction results from several example images under 50 steps. It is evident that DDIM reconstructs images with non-negligible distortions compared to the original images, as marked by the red rectangle in Figure 4. Our findings suggest that the exact inversion samplers (EDICT, BDIA, and O-BELM) indeed achieve exact inversion at the latent level, thereby achieving the lower bound of the reconstruction error of AE in latent diffusion models. Although the encoding and decoding processes of AE introduce some reconstruction error, these errors do not result in any detectable inconsistencies in the image as perceived by the human eye. It's also important to note that exact inversion requires the storage of two intermediates for precise reconstruction. This is feasible in downstream tasks such as image editing.

We have also conducted an additional experiment to assess the reconstruction error in the latent space of O-BELM and other baseline methods as shown in Figure 5.

Table 5: Comparison of different samplers on MSE reconstruction loss on latent space on COCO-14.

| | MSE loss of reconstruction on latents | | | |
| --- | --- | --- | --- | --- |
| | DDIM | EDICT | BDIA | **O-BELM** |
| 10 steps | 0.414 | 0.000 | 0.000 | 0.000 |
| 20 steps | 0.243 | 0.000 | 0.000 | 0.000 |
| 50 steps | 0.063 | 0.000 | 0.000 | 0.000 |
| 100 steps | 0.041 | 0.000 | 0.000 | 0.000 |

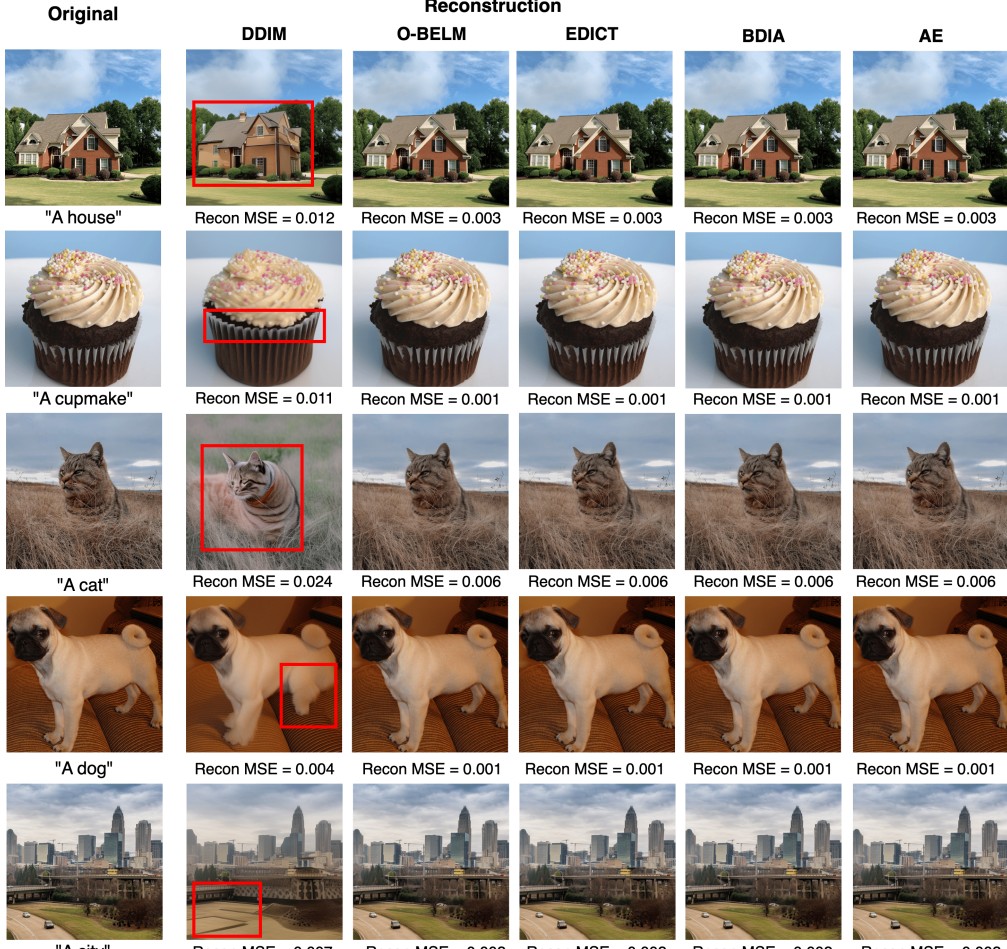

Figure 4: Results of image reconstruction and MSE error using DDIM and exact inversion samplers under 50 steps. The red rectangle point out the inconsistent part in the reconstructed images of DDIM.

## C.2 Image Generation Results

**hyperparameter choosing for EDICT and BDIA**   For EDICT and BDIA, each has an additional hyperparameter ($\gamma$ and $p$ respectively) whose optimal values are sensitive to the task at hand. To ascertain their appropriate hyperparameters for CIFAR-10 and CelebA-HQ, we executed a grid search in the 10-step scenario, as depicted in Table 6 and Table 7. These values were then fixed when performing cases with more steps. We evaluated $\gamma$ in BDIA from 0 to 1 with a grid increment of 0.1, and assessed $p$ in EDICT from 0.90 to 0.97 with a grid increment of 0.01, adhering to their suggested hyperparameter intervals. For the text-guided generation task using COCO-14 captions, we employed the values recommended in their respective papers.

**Guidance Weight in Conditional Generation**   The 30k prompts is randomly selected from the COCO dataset [35] as the test set. For the text-guided generation task, we utilize a classifier-free

Table 6: Comparison of FID score( ↓) of BDIA method for the task of CIFAR10/CelebA-HQ generation with different choice of $\gamma$ in the 10-step scenario.

| BDIA | The choice of $\gamma$ | | | | | | | | | | |
|---|---|---|---|---|---|---|---|---|---|---|---|
| | 0.0 | 0.1 | 0.2 | 0.3 | 0.4 | 0.5 | 0.6 | 0.7 | 0.8 | 0.9 | 1.0 |
| CIFAR10 | 17.41 | **12.27** | 15.60 | 23.62 | 33.39 | 43.93 | 54.93 | 66.32 | 78.33 | 92.16 | 106.37 |
| CelebA-HQ | **27.41** | 29.52 | 41.66 | 52.56 | 61.19 | 68.94 | 76.31 | 83.53 | 91.18 | 98.95 | 106.24 |

Table 7: Comparison of FID score( ↓) of EDICT method for the task of CIFAR10/CelebA-HQ generation with different choice of $p$ in the 10-step scenario.

| EDICT | The choice of $p$ | | | | | | | |
|---|---|---|---|---|---|---|---|---|
| | 0.90 | 0.91 | 0.92 | 0.93 | 0.94 | 0.95 | 0.96 | 0.97 |
| CIFAR10: 10 steps | 149.00 | 142.81 | 135.05 | 127.52 | 119.57 | 110.50 | 99.86 | **87.11** |
| CelebA-HQ: 10 steps | 82.16 | 78.09 | 74.10 | 70.18 | 66.61 | 63.16 | 60.43 | **57.82** |

technique [22] which requires a guidance weight. For BDIA, we select a guidance weight of $4.0$ and for EDICT, we choose $3.0$, as recommended in their respective papers. For DDIM, we perform a grid search in the 20-step scenario, as shown in Table 8, and determine the optimal guidance weight to be $5.5$. This value is then fixed for other scenarios as well as for the O-BELM sampler.

Table 8: Comparison of FID score( ↓) of DDIM method for the task of text-guided generation with different choice of guidance weight.

| DDIM | The choice of guidance weight | | | | | | | | | | |
|---|---|---|---|---|---|---|---|---|---|---|---|
| | 2.5 | 3.0 | 3.5 | 4.0 | 4.5 | 5.0 | 5.5 | 6.0 | 6.5 | 7.0 | 7.5 |
| COCO-14 | 26.03 | 22.66 | 20.90 | 20.14 | 19.66 | 19.47 | **19.45** | 19.47 | 19.61 | 19.81 | 19.97 |

**Examples of O-BELM** We present unconditionally generated samples of O-BELM sampler in Figure 5(a) (CIFAR10, $32 \times 32$) and Figure 5(b) (CelebA-HQ, $256 \times 256$). Furthermore, we display text-guided generated samples in Figure 6, utilizing the pretrained SD-1.5 model [48] ($512 \times 512$) with captions from the COCO-14 dataset [35]. The guidance weight for our O-BELM has been set at $5.5$ to align with the choice of DDIM.

### C.3 Image Interpolation

Image interpolation refers to the process of morphing between two images by interpolating between their corresponding latent vectors in the latent space, usually expecting to achieve a smooth transition between these images.

The diffusion ODE (5) establishes a correspondence between latent noise and samples, which can also be perceived as a coding for the samples. Given that O-BELM can more effectively simulate the diffusion ODE while preserving the one-to-one relationship of the coding, we believe that the exact inversion of O-BELM can intrinsically provide a more rational correspondence. This, in turn, facilitates superior interpolation effects.

We follow the experiment setting in [53] to generate interpolations on a line, which randomly sample two initial values $x_T^{(0)}$ and $x_T^{(1)}$ from the standard Gaussian $\mathcal{N}(0, 1)$, interpolate them with spherical linear interpolation [51], then use the BELM to obtain $x_0$ samples. The spherical linear interpolation $x_T^{(\alpha)}$ is calculated by

$$\mathbf{x}_T^{(\alpha)} = \frac{\sin((1-\alpha)\theta)}{\sin(\theta)}\mathbf{x}_T^{(0)} + \frac{\sin(\alpha\theta)}{\sin(\theta)}\mathbf{x}_T^{(1)}, \tag{64}$$

where $\theta = \arccos\left(\frac{\left(\mathbf{x}_T^{(0)}\right)^T\left(\mathbf{x}_T^{(1)}\right)}{\left\|\mathbf{x}_T^{(0)}\right\|\left\|\mathbf{x}_T^{(1)}\right\|}\right)$. We demonstrate the interpolation results of various models including CelebA-HQ (a), Butterflies (b), Emoji (c) and Anime (d) in Figure 7.

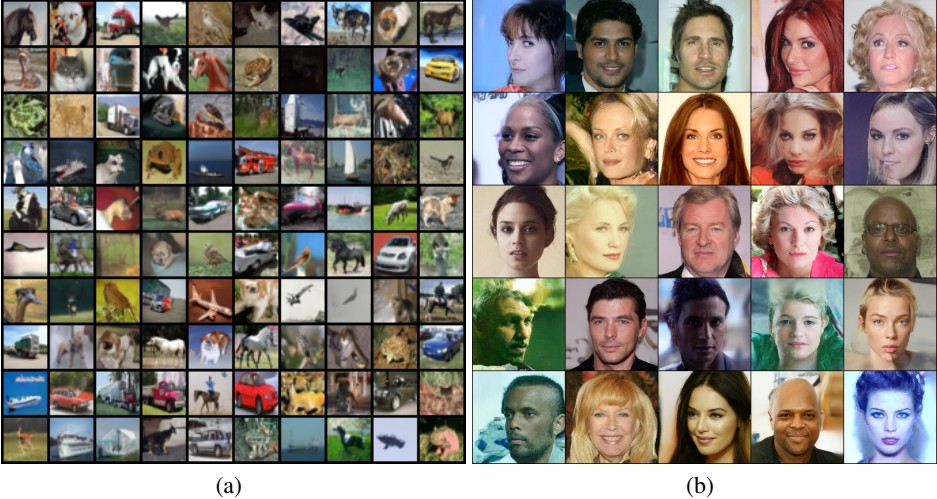

(a)                                                    (b)

Figure 5: (a) uncurated CIFAR10 samples with BELM, steps = 100 (b) uncurated CelebA-HQ samples with BELM, steps = 100

## C.4   Image Editing

We adhere to the experimental setup of [63], initially introducing inversion noise to the images while preserving 20 percent of the steps during the inversion process. We utilize new prompts to reconstruct and edit the images. The guidance weight is consistently set at 3.0 for all instances.

### C.4.1   ControlNet-Based Image Editing

We evaluated O-BELM and baseline algorithms on ControlNet-based image editing tasks, which included canny-based and depth-map-based editing as illustrated in Figure 8. The editing hyperparameters are chosen the same as our original paper. The ControlNet hyperparameters were kept at their default values, consistent across all methods. We set the number of steps to 100. The canny images were obtained using the Canny function from the opencv-python library, and the depth-map model used was Intel/dpt-large (`https://huggingface.co/Intel/dpt-large`). We use stable-diffusion-v1-5 model (`https://huggingface.co/runwayml/stable-diffusion-v1-5`) as our base model.

### C.4.2   Style Transfer

We evaluated O-BELM and baseline algorithms on style transfer tasks using the style transfer sub-dataset of the PIE-Bench dataset (`https://paperswithcode.com/dataset/pie-bench`) as illustrated in Figure 9. The editing hyperparameters were selected to match those in our original paper. We use stable-diffusion-2-base model (`https://huggingface.co/stabilityai/stable-diffusion-2`) as our base model.

## C.5   Pretrained Models

All of the pretrained models used in our research are open-sourced and available online as follows:

- CIFAR10 generation : ddpm_ema_cifar10
  https://github.com/VainF/Diff-Pruning/releases/download/v0.0.1/ddpm_ema_cifar10.zip
- CelebA-HQ generation and interpolation : ddpm-ema-celebahq-256
  https://huggingface.co/google/ddpm-ema-celebahq-256
- Text-to-Image generation : stable-diffusion-v1-5, stable-diffusion-2-base
  https://huggingface.co/runwayml/stable-diffusion-v1-5
  https://huggingface.co/stabilityai/stable-diffusion-2-base

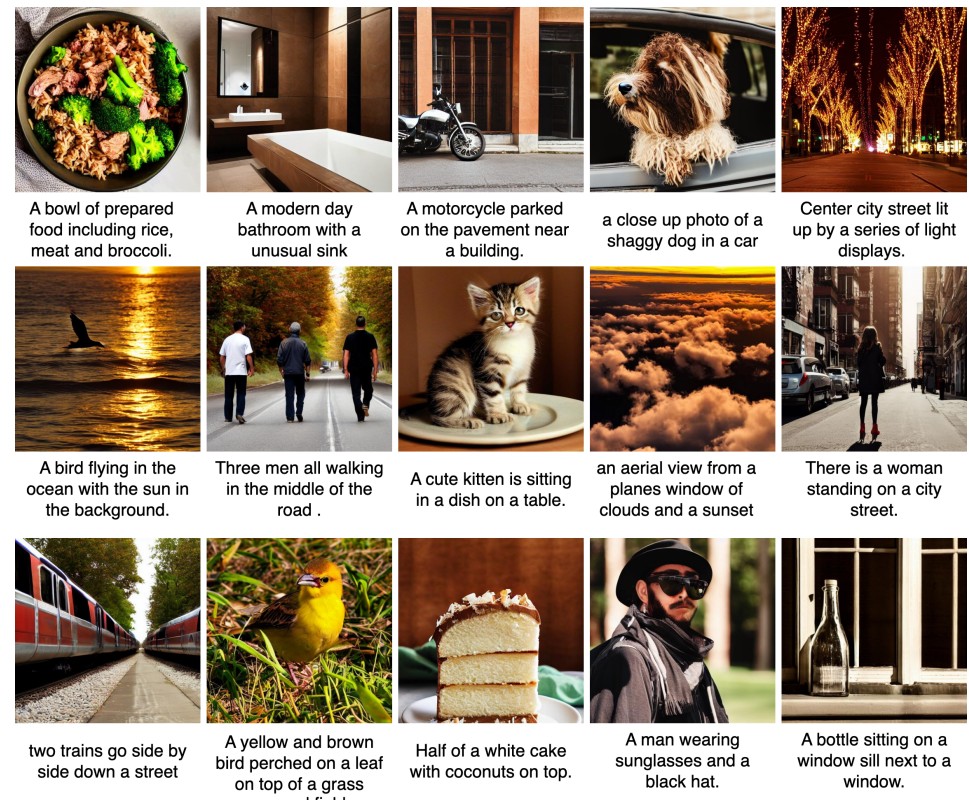

Figure 6: Prompts and generated images by O-BELM on COCO-14 dataset using SD-1.5 with 100 steps.

- Butterflies interpolation : ddim-butterflies-128
  https://huggingface.co/dboshardy/ddim-butterflies-128
- Emoji interpolation : ddpm-EmojiAlignedFaces-64
  https://huggingface.co/Norod78/ddpm-EmojiAlignedFaces-64
- Anime interpolation : ddpm-ema-anime-256
  https://huggingface.co/mrm8488/ddpm-ema-anime-256

**The scheduler setting**   For these pre-trained diffusion models, we adopt the noise scheduler outlined in their respective configurations and apply it consistently across all our experiments. As our experiments do not involve the training or fine-tuning of diffusion models, there is no requirement to develop a new scheduler setting.

# D   Discussions

## D.1   Hyperparameters of BDIA and EDICT

Notice that, the intuitive exact inversion samplers achieve exact diffusion inversion at a cost of introducing an additional hyperparameter. comparing to DDIM, including both BDIA (10) (with additional hyperparameter $\gamma$) and EDICT (9) (with additional hyperparameter $p$). We point out that the need for additional hyperparameters would hinder the widespread application of the exact inversion samplers. The sampling quality of the previous exact inversion samplers is highly inrobust to the additional hyperparameter. EDICT recommend to choose $p \in [0.9, 0.97]$ as EDICT would result in inconvergence when $p \leq 0.9$.

As depicted in Figure 10, we observe that the use of different hyperparameters within the recommended interval could potentially result in divergence. In Table 6 and Table 7, we note that the Frechet

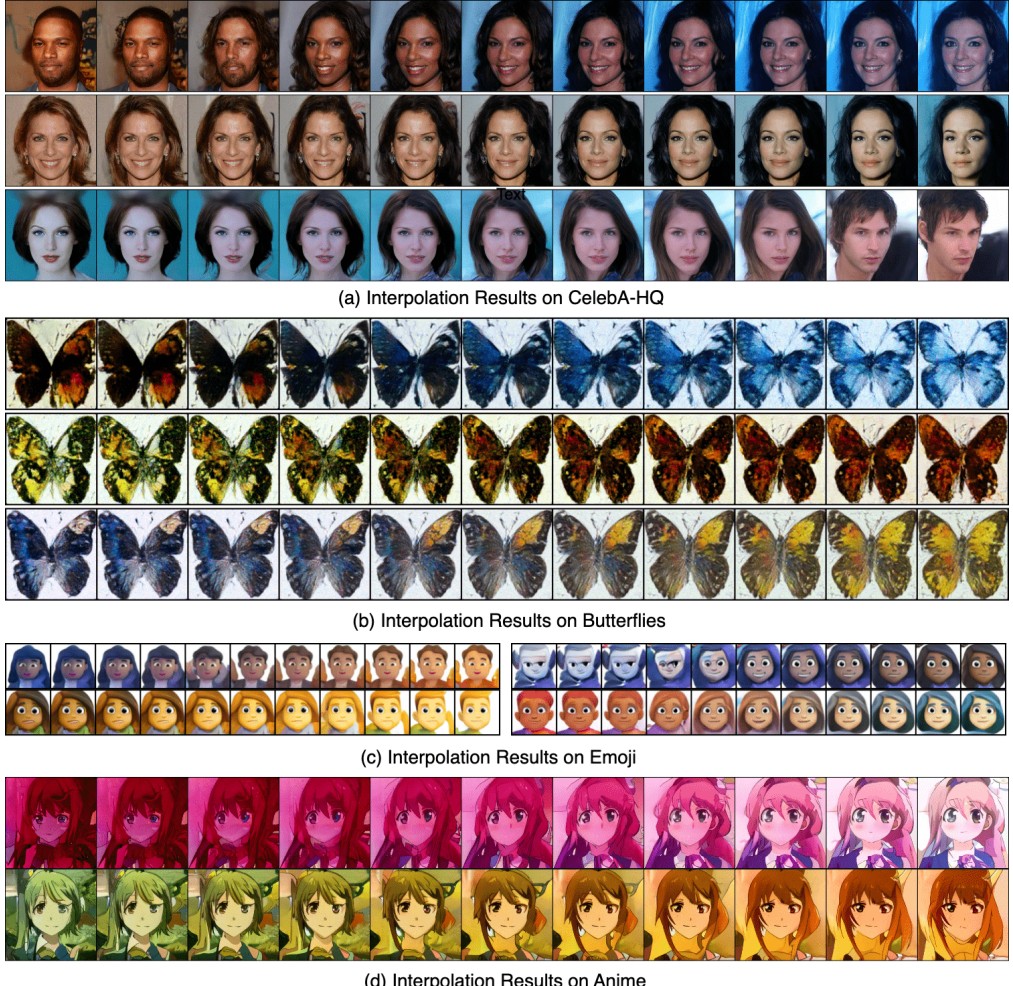

(a) Interpolation Results on CelebA-HQ

(b) Interpolation Results on Butterflies

(c) Interpolation Results on Emoji

(d) Interpolation Results on Anime

Figure 7: Interpolation of samples of various models using O-BELM with 100 steps.

Inception Distance (FID) fluctuates significantly with respect to these unstable hyperparameters. Furthermore, the optimal hyperparameters vary across different datasets and steps.

## D.2 The Different Definition on LTE

We would like to draw our readers' attention to the fact that the term Local Truncation Error (LTE) as used in this paper might differ from its usage in some other mathematical papers [58, p.317(12.24)]. Specifically, what is referred to as $\tau_i/h_i$ in this paper is often called LTE in other contexts, implying that their definition of LTE includes an additional division by a stepsize. However, in the context of variable-stepsize-variable-formula (VSVF), our definition proves to be more convenient and is more commonly adopted in papers dealing with VSVF [8, (2.1)].

## D.3 Time Complexity and Memory Complexity

**Time complexity**  Regarding the sampling task of diffusion models, the time cost bottleneck is the access to the noise network $\varepsilon_\theta(\mathbf{x}_i, i)$. The number of accesses to $\varepsilon_\theta(\mathbf{x}_i, i)$ is also referred to as NFE (the number of function evaluations). For the same value of $N$, we observed that O-BELM, DDIM, and BDIA all require an NFE equal to $N$ for a single sampling chain. However, EDICT doubles this requirement to $2N$.

Experimentally, we've conducted additional tests to compare the average time cost of different methods across sampling, editing, and reconstruction tasks. The results show that O-BELM does

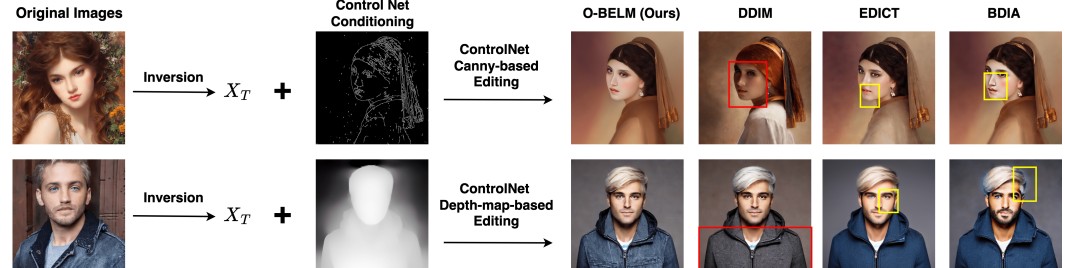

Figure 8: Comparison of ControlNet-based editing results of different samplers. DDIM leads to inconsistencies (red rectangle), and the EDICT and BDIA samplers introduce low-quality sections (yellow rectangle). Our O-BELM sampler ensures consistency and demonstrates high-quality results, even in such large scale editing and still preserve features from original images (face in the first example and clothing in the second example).

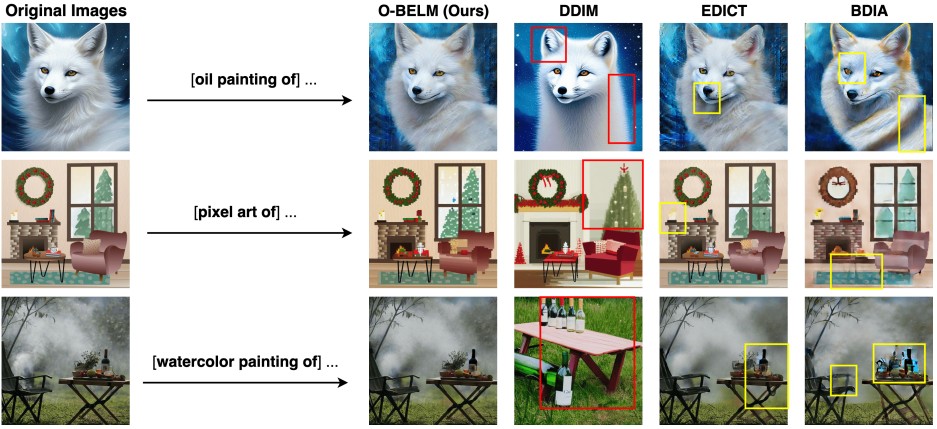

Figure 9: Comparison of Style Transfer results of different samplers on the PIE Benchmark. DDIM leads to inconsistencies (red rectangle), and the EDICT and BDIA samplers introduce low-quality sections (yellow rectangle). Our O-BELM sampler ensures structure preservation and high-quality style transfer, thus show the robustness and effectiveness of O-BELM sampler.

not incur any additional computational overhead compared to DDIM across all these tasks. Detailed information about these experiments can be found in Table 9.

**Memory complexity**   During the sampling process of diffusion models, typically the entire chain of the process is maintained. Both BDIA and O-BELM do not require additional memory beyond the previous sampling path. However, due to the auxiliary states, the memory requirements of EDICT need to be doubled.

All experiments were conducted on a single V100 GPU and an Intel Xeon Platinum 8255C CPU. The sampling of 30k images using SD models under 100 steps took approximately 24 hours. The sampling of 50k images using a pre-trained CIFAR10 model under 100 steps took around 4 hours. Meanwhile, the sampling of 50k images using a pre-trained CelebA-HQ model under 100 steps required about 40 hours.

## D.4   Other Inversion Techniques

We observe that the field of diffusion inversion is rapidly evolving. Recently, several works related to diffusion inversion have been proposed.

For instance, the study by [26] suggests altering the prior distribution, as opposed to using Gaussian noise, for more convenient inversion. However, this approach requires training new models, rendering it incompatible with existing pretrained models.

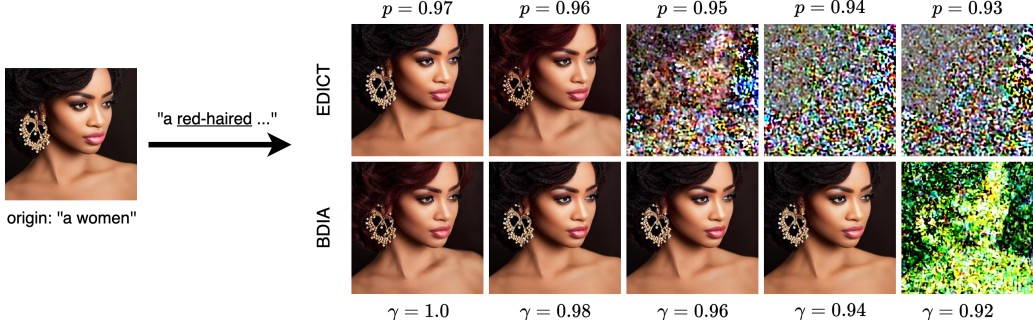

Figure 10: Image editing example for EDICT and BDIA with different hyperparameters, carried out over 200 steps. We observe that even within the interval advised in the original paper, the editing result may still diverge.

| | Time costs of Different Tasks (50 steps) | | |
| --- | --- | --- | --- |
| | Image Generation (s) | Image Editing (s) | Image Reconstruction (s) |
| DDIM | 6.67 | 13.30 | 13.20 |
| EDICT | 12.67 | 25.77 | 25.72 |
| BDIA | 6.59 | 13.37 | 13.28 |
| **O-BELM(Ours)** | **6.53** | **13.22** | **13.20** |

Table 9: Comparison of time costs for different methods on the PIE Benchmark using the SD-2b model, as tested on a single NVIDIA Tesla V100. The results indicate that O-BELM does not incur additional computational time costs compared to DDIM across Generation, Editing, and Reconstruction tasks. We assessed the time costs of O-BELM and baseline algorithms using the PIE-Bench dataset (`https://paperswithcode.com/dataset/pie-bench`), which included tasks such as image generation, image editing, and image reconstruction. The number of steps was set to 50. We employed the stable-diffusion-2-base model (`https://huggingface.co/stabilityai/stable-diffusion-2`) as our base model and conducted tests on a single NVIDIA V100 chip and an Intel Xeon Platinum 8255C CPU.

The research conducted by [72, 16] advocates for the training of a model-dependent bias corrector for precise inversion. Despite this, it fails to achieve mathematically exact inversion.

The work of [39, 23] proposes the use of an implicit method in inversion to align with the sampling. However, this approach is time-consuming and residual optimization errors persist.

And, the study by [34] suggests training a reverse one-step consistency model. However, its experimental performance also demonstrates reconstruction inconsistency.

We understand that there are several techniques proposed to address the inexact inversion issue of DDIM within the context of *classifier-free-text-guided image editing*. These include NMG [5], DirectingInv [28], ProxEdit [17], NPT [41] and NT [42]. We point out that the proposed O-BELM and these techniques should not be considered as comparative algorithms for following reasons.

- These methods are orthogonal. O-BELM modifies the discretization formula to achieve exact inversion, while these techniques adjust the classifier-free-guidance mechanism. They address this problem from different directions.
- They can be used together in the classifier-free-text-guided image editing. Take DirectingInv as an instance, its inversion is just DDIM inversion and its forward process encompasses two state-interacting DDIM forward processes with different prompts. We can substitute the DDIM inversion/forward in DirectingInv to be O-BELM inversion/forward and get O-BELM+DirectingInv.
- Their working scenarios differ. The O-BELM is built on the general diffusion IVP, and can guarantee exact inversion and minimized error under all tasks based on diffusion ODE (PF-ODE). O-BELM can always converge to underlying IVP solution as demonstrated by

Proposition 5. This means that the BELM framework is compatible with a wide variety of diffusion-based tasks, irrespective of the data type (images or words), the task type (editing or interpolation), the guidance method (unconditional, classifier-free, classifier-based, or adjoint ODE-based), or the network structure (whether it includes an attention layer or not). On the contrary, these techniques are developed specific for classifier-free-text-guided image editing task.

## D.5 Broader (Social) Impacts

The development of accurate and stable exact inversion diffusion sampling like O-BELM for DMs, as discussed in this paper, holds significant potential for several domains, including machine learning, healthcare, environmental modeling, and economics.

However, while this research holds great potential for positive impacts, it is also important to consider potential negative societal impacts. The enhanced ability of an accurate and stable exact diffusion sampler could potentially be misused. For instance, it could be exploited to creating deepfakes, leading to misinformation. It may also raise privacy concerns, as more detailed and source data can be decoded from the intermediates using O-BELM. In healthcare, if not properly regulated, the use of synthetic patient data could lead to ethical issues. Therefore, it is crucial to ensure that the findings of this research are applied ethically and responsibly, with necessary safeguards in place to prevent misuse and protect privacy.

## D.6 Limitations

This paper does not explore the integration of high-accuracy exact inversion samplers such as O-BELM with more powerful image editing pipelines. Additionally, the application of high-accuracy exact inversion samplers like O-BELM to tasks beyond image processing remains uninvestigated in this work. The concept of employing bidirectional explicit constraints to ensure exact inversion when applied to accelerated DM-solvers remains unexplored. There is also a lot of downstream tasks of DMs that a exact inversion samplers like O-BELM can apply [64, 61, 60, 4, 66, 13, 59, 70, 15, 14]. It will be also interesting to apply the exact inversion ODE sampler in variational inference [69, 65] or flow matching [36, 76].

