# OpenReview forum: "BELM: Bidirectional Explicit Linear Multi-step Sampler for Exact Inversion in Diffusion Models"
_NeurIPS.cc/2024/Conference — NeurIPS 2024 poster_

### Official Review · Reviewer_HZYw · 2024-07-12

**Soundness:** 3
**Presentation:** 3
**Contribution:** 3
**Rating:** 7
**Confidence:** 4

**Summary:**

This paper introduces a systematic framework, referred to as BELM, that is designed for the specific task of exact inversion of diffusion sampling. This framework encompasses several existing intuitive exact inversion samplers as its special cases. Subsequently, the authors derive an optimal variant within this framework via local truncation error minimization, named O-BELM, and investigate its theoretical properties, including zero-stability and convergence. Experimental results demonstrate that O-BELM can achieve exact inversion and high-quality sampling. The authors further explore its potential applications in downstream computer vision tasks.

**Strengths:**

1 This paper represents the first attempt to formalize the task of exact inversion of diffusion sampling as a rigorous mathematical problem. It highlights that bidirectional explicit property is a sufficient condition for a general sampler to achieve exact inversion.
2 This paper introduces O-BELM as an efficient sampler intended for practical use. Additionally, the paper offers theoretical guarantees for the newly proposed sampler, albeit under mild assumptions.
3 This paper derived the form of local truncation error for general linear multistep samplers within the context of diffusion models.
4 A comprehensive range of experiments has been conducted to confirm that the O-BELM exhibits both exact inversion and high-quality sampling capabilities.
5 This paper is well-articulated and clearly presented.

**Weaknesses:**

1 There is a typographical error in the caption of Table 3.
2 The information regarding the scheduler setting used in the experiments has not been included.
3 Some mathematical derivations in the paper are too succinct, making them difficult to follow. For instance, the transition from equation (30) to equation (31) is not straightforward as it involves a series of Taylor's expansions.

**Questions:**

1 This paper examines the zero-stable property of the O-BELM algorithm and baseline samplers. However, I've noticed that there are other stability properties, such as Absolute-stability or Butcher-stability, for ODE solvers. I'm interested in knowing how the samplers discussed in this paper fare in regards to these stability terms.
2 I'm curious to understand how adopting this new sampler might affect the marginal distribution of the diffusion models. Could you elaborate on the potential impacts this could have?

**Limitations:**

The authors have acknowledged the unexplored semi-linear structure as a limitation in the design of the BELM.
Also, the effects of utilizing O-BELM in advanced image editing like P2P remain unexplored, which is a limitation of O-BELM as well.
Additionally, the impact of the paper is primarily confined to the sampling sub-area within the diffusion-based model community.

---

> ### Author Rebuttal · Authors · 2024-08-01
>
> Thank you for your appreciation of our work, we will answer your questions one by one regarding these weaknesses/questions.
>
>
> > **Weaknesses 1:**  ''There is a typographical error in the caption of Table 3.''
>
> We apologize for the oversight. We have now corrected the error and ensured that the caption accurately reflects the contents of the table. The caption of the Table 3 should be "Comparison of different samplers on FID score( $\downarrow$) for the task of unconditional generation." We have conducted a thorough second revision and fix all typographical errors.
>
>
> > **Weaknesses 2:** ''The information regarding the scheduler setting used in the experiments has not been included.''
>
> We understand the importance of this detail for the complete understanding and replication of our experiment. For these pretrained diffusion models, we utilize the noise scheduler delineated in their respective configurations, applying it uniformly across all our experiments. Given that our experiments do not necessitate the training or fine-tuning of diffusion models, there is no need to devise a new scheduler setting.
> We have now added the necessary information about the scheduler setting in the Appendix C.6 section of the paper.
>
>
> > **Weaknesses 3:** ''Some mathematical derivations in the paper are too succinct, making them difficult to follow. For instance, the transition from equation (30) to equation (31) is not straightforward as it involves a series of Taylor's expansions.''
>
> We acknowledge that some of the transitions, such as from equation (30) to equation (31), may not be easy to follow. We have provided more detailed explanation of these derivations to ensure that the logic and process of our mathematical derivations are clear and easy to follow.
> To elucidate the derivation from Equation (30) to Equation (31), we have included an intermediate illustration as follows:
> $$
> \begin{aligned}
> & \sum\_{j=1}^{k} a\_{i,j} \bar{x}\_{i-1+j} +\sum\_{j=1}^{k-1}b\_{i,j} h\_{i-1+j}\bar{\boldsymbol{\varepsilon}}\_\theta(\bar{x}\_{i-1+j},\bar{\sigma}\_{i-1+j})\\\\
> =&\sum\_{j=1}^{k} a\_{i,j} ( \bar{x}\_{i-1} + \sum\_{l=1}^{2k-2} \frac{1}{(l)!}(\sum\_{m=0}^{j-1}h\_{i+m})^{l}\bar{\boldsymbol{\varepsilon}}^{(l-1)}\_\theta(\bar{x}(t\_{i-1}),\bar{\sigma}\_{i-1}) ) \\\\
> &+\sum\_{j=1}^{k-1}b\_{i,j} h\_{i-1+j}( \sum\_{l=1}^{2k-2} \frac{1}{(l-1)!}(\sum\_{m=0}^{j-1}h\_{i+m})^{l-1}\bar{\boldsymbol{\varepsilon}}^{(l-1)}\_\theta(\bar{x}(t\_{i-1}),\bar{\sigma}\_{i-1}) )\\\\
> &+\mathcal{O}(({\sum\_{m=0}^{k-1}h\_{i+m}})^{(2k-1)})\\\\
> =& \sum\_{j=1}^{k} a\_{i,j} \bar{x}\_{i-1} + \sum\_{l=1}^{2k-2}(\frac{1}{(l!) }\sum\_{j=1}^{k}a\_{i,j}( \sum\_{m=1}^{j}h\_{i+m-1})^{l})\bar{\boldsymbol{\varepsilon}}^{(l-1)}\_\theta(\bar{x}(t\_{i-1}),\bar{\sigma}\_{i-1})\\\\
> &+\sum\_{l=1}^{2k-2}(\frac{1}{((l-1)!) }\sum\_{j=1}^{k-1}b\_{i,j}h\_{i+j-1}( \sum\_{m=1}^{j}h\_{i+m-1})^{l-1})\bar{\boldsymbol{\varepsilon}}^{(l-1)}\_\theta(\bar{x}(t\_{i-1}),\bar{\sigma}\_{i-1})\\\\
> &+\mathcal{O}(({\sum\_{m=0}^{k-1}h\_{i+m}})^{(2k-1)})
> \end{aligned}
> $$
> We have also conducted a comprehensive review of our mathematical derivations, enhancing them with additional details to improve their clarity and ease of understanding.
>
>
>
> > **Questions 1:** ''This paper examines the zero-stable property of the O-BELM algorithm and baseline samplers. However, I've noticed that there are other stability properties, such as Absolute-stability or Butcher-stability, for ODE solvers. I'm interested in knowing how the samplers discussed in this paper fare in regards to these stability terms.''
>
> Zero-stability and Absolute-stability (abbreviated as A-stability) are the most commonly used stability conditions. The A-stability is quite similar to zero-stability, with the primary difference being that A-stability measures the discrete step-size case, while zero-stability measures the limiting step-size case. However, it has been demonstrated that A-stability is extremely demanding and no explicit linear multistep method can satisfy A-stability as shown by the Second Barrier Theorem of Dahlquist [1]. Given the heavy cost of using implicit methods in the diffusion model area, A-stability is not suitable for analysis in the diffusion sampler.
>
> On the other hand, Butcher-stability (abbreviated as B-stability) is typically used to analyze the stability property of Runge-Kutta type ODE samplers. For these samplers, B-stability can derive zero-stability [2]. However, for non-RK type ODE samplers, the implications of the Butcher-stability condition remain unclear.
>
> Therefore, it is reasonable to select zero-stability as the criterion for assessing stability in the analysis of diffusion samplers.
>
>
> > **Questions 2:** ''I'm curious to understand how adopting this new sampler might affect the marginal distribution of the diffusion models. Could you elaborate on the potential impacts this could have?''
>
> The marginal distribution of the proposed sampler will also conform to the marginal distribution implied by the diffusion ODE.
> Intuitively, the approximated $x_t$ obtained from the discretized sampler serves as a proxy for the underlying truth $\mathbf{x}(t)$ of the diffusion initial value problem (IVP).
> Theoretically, our convergence analysis ensures that our approximated $\mathbf{x}_t$ will indeed converge to this underlying truth. The DDIM and other deterministic diffusion samplers, such as the DPM-solver, are also constructed to simulate this diffusion IVP problem. Their marginal distributions are identical, when omitting discretization errors.
> Experimental results show that our O-BELM can generate high-quality samples, effectively demonstrating the capability of O-BELM to accurately model the underlying data distribution at time $T$.
>
> **References**
>
> [1] G. Dahlquist, A special stability problem for linear multistep methods, BIT 3, 27–43, 1963.
>
> [2] Theorem 357D, Butcher, John Charles. Numerical methods for ordinary differential equations. John Wiley & Sons, 2016.

---

> > ### Comment · Reviewer_HZYw · 2024-08-14
> >
> > After reading the careful response and other reviews, I will keep my score. Thank the authors for their response.

---

> > > ### Author Response · Authors · 2024-08-14
> > >
> > > Thanks for your valuable questions regarding our theory and for your generous appreciations to our work!

---

### Official Review · Reviewer_eBdS · 2024-07-12

**Soundness:** 2
**Presentation:** 3
**Contribution:** 2
**Rating:** 6
**Confidence:** 5

**Summary:**

This paper introduces a novel method for inverting real images into a diffusion model. It presents Bidirectional Explicit Linear Multi-step (BELM) samplers aimed at minimizing the mismatch in DDIM inversion.

**Strengths:**

The proposed Bidirectional Explicit Linear Multi-step (BELM) samplers seems reasonable, and outputforms DDIM sampler.

**Weaknesses:**

1. The paper lacks discussion and comparison with several related works such as NMG, EDICT, DirectingInv, ProEdit, ReNoise, and others. These works also aim to address the limitations of DDIM inversion through various proposed solutions. It is essential for the authors to provide a thorough discussion and comparative analysis with these existing methods.

2. The evaluation metrics used in the paper may not be sufficiently convincing. DirectingInv, for example, introduces several metrics specifically tailored for evaluating editing tasks, which are generally considered more reliable. Furthermore, DirectingInv also establishes a standard benchmark dataset for diffusion inversion tasks. It would be beneficial for the authors to conduct experiments using this benchmark dataset to ensure a comprehensive study and rigorous evaluation of their proposed method.

3. In terms of reconstruction performance compared to AE, EDICT, and BDIA, O-BELM does not demonstrate any improvement.

**Questions:**

Please see in weakness

---

> ### Author Rebuttal · Authors · 2024-08-06
>
> Thank you for your valuable reviews, we will answer your questions one by one regarding these weaknesses/questions.
>
> > **Weaknesses 1:** The paper lacks discussion and comparison with several related works such as NMG, EDICT, DirectingInv, ProEdit, ReNoise, and others. These works also aim to address the limitations of DDIM inversion through various proposed solutions. ...
>
> We understand that there are several techniques proposed to address the inexact inversion issue of DDIM within the context of **classifier-free-text-guided image editing**. These include the methods you mentioned, such as NMG[1], DirectingInv[2], ProEdit[3], and ReNoise[4], as well as others like NPT [5] or NT [6]. We argue that the proposed O-BELM and these techniques should not be considered as comparative algorithms for following reasons.
> - **These methods are orthogonal.** O-BELM modifies the discretization formula to achieve exact inversion, while these techniques adjust the classifier-free-guidance mechanism. They address this problem from different directions.
> - **They can be used together in the classifier-free-text-guided image editing.** Take DirectingInv as an instance, its inversion is just DDIM inversion and its forward process encompasses two state-interacting DDIM forward processes with different prompts. We can substitute the DDIM inversion/forward in DirectingInv to be O-BELM inversion/forward and get O-BELM+DirectingInv. We conduct experiments of origin DDIM+DirectingInv and O-BELM+DirectingInv in `Table 1 in attached PDF`, showing that O-BELM can be integrated with DirectingInv to further enhance the performance.
> - **Their working scenarios differ.** The O-BELM is built on the general diffusion IVP (equation 11), and can guarantee exact inversion (Proposition 2) and minimized error (Proposition 4) under all tasks based on diffusion ODE (PF-ODE). O-BELM can always converge to underlying IVP solution as demonstrated by Proposition 5. This means that the BELM framework is compatible with a wide variety of diffusion-based tasks, irrespective of the data type (images or words), the task type (editing or interpolation), the guidance method (unconditional, classifier-free, classifier-based, or adjoint ODE-based), or the network structure (whether it includes an attention layer or not). On the contrary, these techniques are developed specific for classifier-free-text-guided image editing task.
>
> We will add a paragraph to discussion these works.
>
> > **Weaknesses 2:** The evaluation metrics used in the paper may not be sufficiently convincing. DirectingInv, for example, introduces several metrics specifically tailored for evaluating editing tasks, which are generally considered more reliable. Furthermore, DirectingInv also establishes a standard benchmark dataset for diffusion inversion tasks. ...
>
> Although we have conducted thorough theoretical analysis on BELM framework and use experiments as validation to our theory, we agree that comprehensive evaluation metrics on standard benchmark dataset is beneficial. To illustrate the efficiency of our proposed O-BELM at downstream image editing task, we follow your valuable advice to compare O-BELM with baseline methods on the standard benchmark PIE-Bench established by DirectingInv [2]. We follow the experiment design of DirectingInv to evaluate in terms of eight metrics covering four aspects:  structure distance, background preservation (PSNR, LPIPS, MSE, and SSIM outside the annotated editing mask), edit prompt-image consistency (CLIPSIM of the whole image and regions in the editing mask) and inference time. These results can be found in `Table 1 in the attached PDF` and `Table 2 in the Author Rebuttal`. It is shown that O-BELM outperform baselines in this benchmark.
>
>
> > **Weaknesses 3:** In terms of reconstruction performance compared to AE, EDICT, and BDIA, O-BELM does not demonstrate any improvement.
>
> This is because O-BELM, BDIA, and EDICT all fall within the BELM framework, and as Proposition 2 proves, any BELM method possesses the exact inversion property. However, since SD-1.5 is a latent diffusion model (LDM), errors will be inevitably introduced during the encoding/decoding process of the AutoEncoder (AE) component of SD model. It is a prevailing practice to take the reconstruction error of AE as a lower bound of LDM pixel-level reconstruction error[7]. O-BELM has already reached this lower bound.
>
> We have conducted an additional experiment to assess the reconstruction error in the latent space of O-BELM and other baseline methods.
> | Method|  MSE on latents (10 steps) | MSE on latents (20 steps)    | MSE on latents (100 steps)  |
> | :--| :- | :-|:-|
> |DDIM|0.414|0.243|0.041|
> |EDICT|0.000|0.000|0.000|
> |BDIA|0.000|0.000|0.000|
> |**O-BELM (Ours)**|**0.000**|**0.000** |**0.000**|
>
> It is demonstrated that O-BELM (along with BDIA and EDICT) achieves exact inversion at the latent level, resulting in zero reconstruction error. O-BELM outperforms BDIA and EDICT in terms of sampling accuracy due to reduced local error (as shown in Table 1 of the original paper), not in terms of reconstruction error.
>
> **References**
>
> [1] Cho, Hansam, et al. "Noise map guidance: Inversion with spatial context for real image editing." arXiv:2402.04625 (2024).
>
> [2] Ju, Xuan, et al. "Direct inversion: Boosting diffusion-based editing with 3 lines of code." arXiv:2310.01506 (2023).
>
> [3] Han, Ligong, et al. "Proxedit: Improving tuning-free real image editing with proximal guidance." WACV 2024.
>
> [4] Garibi, Daniel, et al. "ReNoise: Real Image Inversion Through Iterative Noising." arXiv:2403.14602 (2024).
>
> [5] Miyake, Daiki, et al. "Negative-prompt inversion: Fast image inversion for editing with text-guided diffusion models." arXiv:2305.16807 (2023).
>
> [6] Mokady, Ron, et al. "Null-text inversion for editing real images using guided diffusion models." CVPR 2023.
>
> [7] Wallace, Bram, Akash Gokul, and Nikhil Naik. "Edict: Exact diffusion inversion via coupled transformations." CVPR 2023.

---

> ### Author Response · Authors · 2024-08-13
> **Looking forward for further discussions.**
>
> Dear Reviewer [eBdS],
>
> Thank you again for your constructive feedback.
>
> Our research is the first to construct a theoretical well-posed IVP modeling for the general inversion problem in diffusion sampling, as outlined in **Equation 11** and **Proposition 1**. Based on this IVP view, we have innovatively identified the Bidirectional Explicit condition, a sufficient prerequisite for achieving a mathematically precise inversion, as stated in **Proposition 2**. This condition does not merely reduce inversion error, but ensures exactness. Building on this condition, we have developed a generic formula for the general exact inversion samplers, which we have termed as Bidirectional Explicit Linear Multi-step (BELM) samplers, as detailed in **Equation 14**. These samplers incorporate several previous methods as special cases, as noted in **Remark 1**. We have conducted a thorough analysis of the Local Truncation Error (LTE) within the BELM framework, as described in **Proposition 3** and **Corollary 1**, and have proposed the optimal variants, O-BELM, as delineated in **Proposition 4**. We have also demonstrated that O-BELM possesses the advantageous property of zero-stability, as outlined in **Proposition 5**, which ensures its robustness to initial values. Additionally, O-BELM exhibits the beneficial characteristic of global convergence, also stated in **Proposition 5**, which prevents O-BELM from diverging during sampling.
>
> In regard to downstream tasks, we have taken your insightful advice into account and conducted experiments on the **standard image editing benchmark**, **PIE-Bench**. Additionally, during our interactions with Reviewer [QGyL] and Reviewer [brf4], we have conducted further experiments to provide support for the effectiveness and robustness of our methods. These additional tests include the **ControlNet-based editing task** and the **style transfer task**, further showcasing the effectiveness and robustness of our approach.
>
>
> We sincerely hope to engage in further discussions with you. Thank you for your time and consideration!

---

> ### Author Response · Authors · 2024-08-14
> **Dear Reviewer [eBdS], we are eager for your feedback**
>
> Dear Reviewer [eBdS],
>
> Since the discussion period will end in a few hours, we will be online waiting for your feedback on our rebuttal, which we believe has addressed your concerns. We would highly appreciate it if you could take into account our response when having discussions with AC and other reviewers.
>
> Thank you so much for your time and efforts.  Sorry for our repetitive messages, but we're eager to your feedback.
>
> Authors of Submission 2649

---

### Official Review · Reviewer_brf4 · 2024-07-12

**Soundness:** 3
**Presentation:** 3
**Contribution:** 3
**Rating:** 6
**Confidence:** 3

**Summary:**

The paper introduces the Bidirectional Explicit Linear Multi-step (BELM) sampler framework for exact inversion in diffusion models. The authors systematically investigate the Local Truncation Error (LTE) within the BELM framework and propose an optimal variant, O-BELM, which minimizes LTE for high sampling quality. Comprehensive experiments validate O-BELM's effectiveness in tasks like image reconstruction, editing, and interpolation.

**Strengths:**

Novel Framework: The BELM framework generalizes existing exact inversion samplers and introduces a bidirectional explicit constraint to ensure exact inversion.

Theoretical Rigor: The paper provides a thorough theoretical analysis of the Local Truncation Error (LTE) and the stability and convergence properties of the proposed samplers.

Practical Applications: Demonstrates the practical potential of O-BELM in various tasks such as image reconstruction, editing, and interpolation.

**Weaknesses:**

1.	As for downstream applications of diffusion inversion, style transfer [1] should also be included. I encourage the author could apply this method to some style transfer to show the robustness and effectiveness of this framework.
2.	Many punctuation marks at the end of the formulas are omitted by the authors. It is suggested to careful revise this article.
3.	The comparative analysis of qualitative and quantitative results is deficient. It is suggested that the authors provide detail comparative analysis and conclusion in experiment section.

**Questions:**

See weakness.

**Limitations:**

This paper does not explore the integration of high-accuracy exact inversion samplers such as O-BELM with more powerful image editing pipelines. Additionally, the application of high-accuracy exact inversion samplers like O-BELM to tasks beyond image processing remains uninvestigated in this work.

---

> ### Author Rebuttal · Authors · 2024-08-07
>
> Thank you for your valuable reviews, we will answer your questions one by one regarding these weaknesses/questions.
>
> > **Answer to Weaknesses 1:** ''As for downstream applications of diffusion inversion, style transfer [1] should also be included. I encourage the author could apply this method to some style transfer to show the robustness and effectiveness of this framework.''
>
> Despite we have conducted thorough theoretical analysis on the BELM framework and use experiments on downstream tasks as validation to our theory. We agree that more downstream applications of diffusion inversion, like style transfer, is beneficial. Therefore, we have evaluated the O-BELM and baseline methods on the style transfer sub-dataset in the PIE-Bench dataset [1].
> - The qualitative comparative analysis can be found in `Figure 2 in Attached PDF`. The results demonstrate that the O-BELM sampler ensures consistency and produces high-quality results in the style transfer task.
> - We also carry out a quantitative analysis on PIE-Bench style transfer sub-dataset. We employ Structure Distance [2] to evaluate whether the edited image preserves the structure of the original image during style transfer and use CLIP Similarity [3] to measure whether the edited image accurately reflects the meaning of target prompts. The following table shows that O-BELM achieves superior structure preservation according to the Structure Distance metric, validating that our O-BELM achieves the exact inversion property (as per Proposition 2 of the original paper). It is also demonstrated that O-BELM achieves better CLIP Similarity, validating that O-BELM reduces sampling error (as per Proposition 4 of the original paper).
>
> | Style Transfer Methods              | Structure Distance$\times10^3$ ($\downarrow$) | CLIP Similariy ($\uparrow$) |
> | :---------------- | :------ | :---- |
> | DDIM       |   71.1   | 24.82 |
> | EDICT      |   19.5   | 24.39 |
> | BDIA       |  25.3   | 23.20 |
> | **O-BELM (Ours)** |  **18.0**   | **25.09** |
>
> *The details of the experiment can be found in the Global Rebuttal section.*
>
> > **Answer to Weaknesses 2:** ''Many punctuation marks at the end of the formulas are omitted by the authors. It is suggested to careful revise this article.''
>
> Thank you for your detailed review. We have conducted a thorough second revision and corrected all typographical errors. We assure you that all necessary punctuation marks will be included in the formulas in our revised version.
>
>
> > **Answer to Weaknesses 3:** ''The comparative analysis of qualitative and quantitative results is deficient. It is suggested that the authors provide detail comparative analysis and conclusion in experiment section.''
>
> Our theoretical investigation and comparison are thorough and precise. We not only establish the exact inversion property (Proposition 2 of the original paper), local error property (Proposition 4 of the original paper), stability property, and global convergence property (Proposition 5 of the original paper) of O-BELM, but also scrutinize the theoretical properties of other methods. This comprehensive analysis allows us to provide a rigorous theoretical comparison of different samplers in Table 1 of the original paper.
>
> We understand that more comparative analysis of qualitative and quantitative results in downsteam task is still beneficial. Therefore, we not only conduct qualitative and quantitative comparative experiments in style-transfer tasks as mentioned above, but also carry out a thorough quantitative comparison of experiments on image editing tasks. We follow the experimental design of DirectingInv [1] to evaluate methods on the PIE-Bench image editing dataset using eight metrics covering four aspects:  structure distance [2], background preservation (PSNR, LPIPS [4], MSE, and SSIM [5] outside the annotated editing mask), edit prompt-image consistency (CLIPSIM [3] of the whole image and regions in the editing mask) and time cost. These results can be found in `Table 1 in the attached PDF` and `Table 2 in the attached PDF`. It is shown that O-BELM outperform baselines in this benchmark.
>
> **References**
>
> [1] Ju, Xuan, et al. "Direct inversion: Boosting diffusion-based editing with 3 lines of code." arXiv preprint arXiv:2310.01506 (2023).
>
> [2] Tumanyan, Narek, et al. "Splicing vit features for semantic appearance transfer." Proceedings of the IEEE/CVF Conference on Computer Vision and Pattern Recognition. 2022.
>
> [3] Wu, Chenfei, et al. "Godiva: Generating open-domain videos from natural descriptions." arXiv preprint arXiv:2104.14806 (2021).
>
> [4] Zhang, Richard, et al. "The unreasonable effectiveness of deep features as a perceptual metric." Proceedings of the IEEE conference on computer vision and pattern recognition. 2018.
>
> [5] Wang, Zhou, et al. "Image quality assessment: from error visibility to structural similarity." IEEE transactions on image processing 13.4 (2004): 600-612.

---

> > ### Comment · Reviewer_brf4 · 2024-08-12
> > **Official Comment by Reviewer brf4**
> >
> > Thank you for your detailed reply. It addresses most of my concerns.
> >
> > I will keep my rating to weak accept.

---

> ### Author Response · Authors · 2024-08-14
>
> Thank you again for your times and efforts, your suggestions have been very helpful for us!

---

### Official Review · Reviewer_QGyL · 2024-07-13

**Soundness:** 3
**Presentation:** 4
**Contribution:** 3
**Rating:** 7
**Confidence:** 3

**Summary:**

This manuscript introduces a generic formulation named ``Bidirectional Explicit Linear Multi-step'' (BELM) samplers for exact inversion of diffusion sampling. In contrast to DDIM Inversion, BELM inversion establishes the relationship between $x_{i-1}$, $x_i$, $x_{i+1}$, and $\epsilon_\theta(x_i, i)$. The authors prove that BELM is the generic version of EDICT and BDIA.

**Strengths:**

1. The authors propose a new variable-stepsize-variable-formula (VSVF) linear multi-step scheme for exact inversion.
2. The paper investigates the Local Truncation Error (LTE) within the BELM framework.
3. O-BELM is designed by minimizing LTE to ensure minimized local error, and the experiments on COCO validate the exact inversion property of O-BELM.

**Weaknesses:**

1. How about the computation cost and latency of 2-step O-BELM compared with BDIA and EDICT?
2. Lack of more applications such as ControlNet-based Image Editing exemplars, and more failure cases is better for analyzing the limitation of BELM.
3. The local error of O-BELM seems higher than BDIA. Besides, for BDIA, $\gamma$ is tuned for different quality and effect, when $\gamma=1$, BDIA seems zero-stable and global convergence.

**Questions:**

Please see the weakness part.

**Limitations:**

Yes.

---

> ### Author Rebuttal · Authors · 2024-08-07
>
> Thank you for your valuable feedback, we will answer your questions one by one regarding these weaknesses/questions.
>
> > **Answer to Weaknesses 1:** ''How about the computation cost and latency of 2-step O-BELM compared with BDIA and EDICT?''
>
>
> - Theoretically, the computation cost bottleneck of diffusion sampling is the number of accesses to the noise network, $\boldsymbol{\varepsilon}_\theta(\mathbf{x}_i,i)$, which is also referred to as the Number of Function Evaluations (NFE).  In each iteration, O-BELM only accesses the noise network once, as demonstrated in equation 18. Therefore, for a number of steps equal to $N$, O-BELM requires an NFE equal to $N$, which is the same as DDIM and BDIA. However, EDICT doubles this requirement to $2N$.
> Moreover, O-BELM only requires the additional storage of a state, which is a relatively small extra memory cost compared to methods that need to store the entire chain, such as NMG [1] or P2P [2]. And as there is no parallelism in diffusion sampling, the latency is equivalent to the computational time cost.
> - Experimentally, we've conducted additional tests to compare the average time cost of different methods across sampling, editing, and reconstruction tasks. The results show that O-BELM does not incur any additional computational overhead compared to DDIM across all these tasks. Detailed information about these experiments can be found in the `Table 2 in attached PDF` and `Author Rebuttal`.
>
> | Method              |  Time-cost (s) of Image Sampling | Time-cost (s) of Image Editing  |Time-cost (s) of Image Reconstruction  |
> | :---------------- | :------ | :---- |:---- |
> | DDIM       |   6.67   | 13.30 | 13.20 |
> | EDICT      |   12.67   | 25.77 | 25.72 |
> | BDIA       |  6.59   | 13.37 |  13.28|
> | **O-BELM (Ours)** |  **6.53**   | **13.22** | **13.20** |
>
> *The details of the experiment can be found in the Global Rebuttal section.*
>
> > **Answer to Weaknesses 2:** ''Lack of more applications such as ControlNet-based Image Editing exemplars, and more failure cases is better for analyzing the limitation of BELM.''
> - **ControlNet-based Image Editing exemplars:** Despite we have conducted thorough theoretical analysis on the BELM framework and use experiments as validation to our theory. We agree that practical applications such as ControlNet-based Image Editing examples are beneficial for demonstrating the robustness of our O-BELM. Our BELM framework is built on the general diffusion IVP (equation 11), and can guarantee exact inversion (Proposition 2) and minimized error (Proposition 4) under all tasks based on diffusion ODE (PF-ODE). This means that the BELM framework is compatible with a wide variety of diffusion-based tasks, irrespective of the data type (such as image or language), the guidance method (unconditional, classifier-free, classifier-based, ControlNet-based, or adjoint ODE-based), or the network structure (whether it includes an attention layer or not). Following your valuable advice, we conducted experiments on ControlNet-based Image Editing using O-BELM and baseline methods, as shown in `Figure 1 in the Attached PDF`. The results demonstrate that O-BELM is effective in ControlNet-based Image Editing, preserving features from the original images while maintaining high-quality samples.
> - **failure cases:** Compared to EDICT and BDIA, one major advantage of O-BELM is its ability to prevent hard-to-tune hyperparameters and reduce the occurrence of failure cases, as demonstrated in Figure 8 of the original paper. However, we recognize the importance of analyzing failure cases and will include this in our revised version.
>
> > **Answer to Weaknesses 3:** ''The local error of O-BELM seems higher than BDIA. Besides, for BDIA, $\gamma$ is tuned for different quality and effect, when $\gamma=1$, BDIA seems zero-stable and global convergence.''
>
> - **The local error:** Our analysis, as outlined in Corollary 1 and Proposition 4, demonstrates that the asymptotic convergence rate of the local truncation error (LTE) for O-BELM is of order 3 with respect to the small variable $h$, whereas for BDIA it is of order 2. In mathematical terms, a larger order number signifies faster convergence to zero and smaller error [3]. Therefore, the local error of O-BELM is smaller than that of BDIA.
>
> - **zero-stable and global convergence:** The zero-stable property and global convergence property are rigorous and strict mathematical properties, serving as sufficient conditions for achieving stability and high accuracy in practical tasks. While BDIA performs well with some datasets for a certain number of steps when $\gamma = 1$, there is no evidence to suggest that BDIA satisfies these stringent properties. In fact, in Figure 6 of BDIA's original paper [4], BDIA with $\gamma = 1$ could lead to deformed edited images. In Table 5 of our original paper, BDIA with $\gamma = 1$ could lead to low-quality sampling results on CIFAR-10 and CalebA-HQ datasets (FID $\approx 100$).  Furthermore, there is no solid report indicating that the $\gamma$ of BDIA has a clear high-level meaning.
>
> **References**
>
> [1] Cho, Hansam, et al. "Noise map guidance: Inversion with spatial context for real image editing." arXiv:2402.04625 (2024).
>
> [2] Hertz, Amir, et al. "Prompt-to-prompt image editing with cross attention control." arXiv preprint arXiv:2208.01626 (2022).
>
> [3] Rudin, Walter. Principles of mathematical analysis. Vol. 3. New York: McGraw-hill, 1964.
>
> [4] Zhang, Guoqiang, Jonathan P. Lewis, and W. Bastiaan Kleijn. "Exact diffusion inversion via bi-directional integration approximation." arXiv preprint arXiv:2307.10829 (2023).

---

> > ### Comment · Reviewer_QGyL · 2024-08-12
> >
> > Thank you for your rebuttal, my concern about experimental results has been well addressed.
> >
> > After reading the other comments and the corresponding rebuttal, I prefer to raise my score to accept.

---

> ### Author Response · Authors · 2024-08-14
>
> Thank you again for your patience and kindness, your suggestions on experiments have been very helpful!

---

### Author Rebuttal · Authors · 2024-08-07

## To All Reviewers
We sincerely appreciate the time and effort you have dedicated to reviewing our paper! Your valuable feedbacks have been carefully considered, and we have provided point-to-point responses to your reviews in respective rebuttals. We remain open to any additional feedback you may have. If you feel it is appropriate based on our responses, we would be extremely grateful if you could consider raising our score.


## Attached PDF
In the attached PDF document, we have provided several figures and tables. The details of the experiments are listed as follows:

- **Figure 1 -- ControlNet-based Editing Results**
We evaluated O-BELM and baseline algorithms on ControlNet-based image editing tasks, which included canny-based and depth-map-based editing. The editing hyperparameters are chosen the same as our original paper. The ControlNet hyperparameters were kept at their default values, consistent across all methods. We set the number of steps to 100. The canny images were obtained using the Canny function from the opencv-python library, and the depth-map model used was Intel/dpt-large (https://huggingface.co/Intel/dpt-large). We use stable-diffusion-v1-5 model (https://huggingface.co/runwayml/stable-diffusion-v1-5) as our base model.


- **Figure 2 -- Style Transfer Results**
We evaluated O-BELM and baseline algorithms on style transfer tasks using the style transfer sub-dataset of the PIE-Bench dataset (https://paperswithcode.com/dataset/pie-bench). The editing hyperparameters were selected to match those in our original paper. We use stable-diffusion-2-base model (https://huggingface.co/stabilityai/stable-diffusion-2) as our base model.


- **Table 1 -- Quantitative Evaluation in Image Editing on PIE Benchmark**
We evaluated O-BELM and baseline algorithms on image editing tasks using the PIE-Bench dataset (https://paperswithcode.com/dataset/pie-bench). The editing hyperparameters were selected to match those in our original paper. We use stable-diffusion-2-base model (https://huggingface.co/stabilityai/stable-diffusion-2) as our base model. We follow the experimental design of DirectingInv [1] to evaluate methods on the PIE-Bench image editing dataset using seven metrics covering three aspects:  structure distance [2], background preservation (PSNR, LPIPS [3], MSE, and SSIM [4] outside the annotated editing mask), edit prompt-image consistency (CLIPSIM [5] of the whole image and regions in the editing mask). We utilize the dino_vitb8 model (https://huggingface.co/facebook/dino-vitb8) and clip-vit-large-patch14 (https://huggingface.co/openai/clip-vit-large-patch14) for metric evaluations.


-  **Table 2 -- Time Costs Comparison on PIE Benchmark**
We assessed the time costs of O-BELM and baseline algorithms using the PIE-Bench dataset (https://paperswithcode.com/dataset/pie-bench), which included tasks such as image generation, image editing, and image reconstruction. The number of steps was set to 50. We employed the stable-diffusion-2-base model (https://huggingface.co/stabilityai/stable-diffusion-2) as our base model and conducted tests on a single NVIDIA V100 chip and an Intel Xeon Platinum 8255C CPU.




**Reference**

[1] Ju, Xuan, et al. "Direct inversion: Boosting diffusion-based editing with 3 lines of code." arXiv preprint arXiv:2310.01506 (2023).

[2] Tumanyan, Narek, et al. "Splicing vit features for semantic appearance transfer." Proceedings of the IEEE/CVF Conference on Computer Vision and Pattern Recognition. 2022.

[3] Zhang, Richard, et al. "The unreasonable effectiveness of deep features as a perceptual metric." Proceedings of the IEEE conference on computer vision and pattern recognition. 2018.

[4] Wang, Zhou, et al. "Image quality assessment: from error visibility to structural similarity." IEEE transactions on image processing 13.4 (2004): 600-612.

[5] Wu, Chenfei, et al. "Godiva: Generating open-domain videos from natural descriptions." arXiv preprint arXiv:2104.14806 (2021).

---

### Author Response · Authors · 2024-08-14
**Concluding Response**

Dear Reviewers, AC, SAC, and PC,

We would like to begin by expressing our sincere gratitude for your engagement throughout the rebuttal process, which has significantly enhanced the quality of our paper.

In response to reviewers, we have revised the following contents in our manuscript:

- Typographical: The typographical errors in punctuation marks and captions have been corrected, as suggested by Reviewers [brf4] and [HZYw].

- Detailed Mathematical Derivation: A more comprehensive math derivation has been provided, as suggested by Reviewer [HZYw].

- Additional Experiments: In response to the advice of Reviewers [QGyL], [brf4], and [eBdS], we have incorporated further qualitative and quantitative comparative experiments in style-transfer, ControlNet-based image editing, and standard benchmark image editing to demonstrate the effectiveness of O-BELM.

- Clarification: We have incorporated comprehensive explanations for several elements, including time costs (Reviewer [QGyL]), BDIA performance (Reviewer [QGyL]), reconstruction error (Reviewer [eBdS]), and time scheduler (Reviewer [HZYw]).

We are immensely grateful for your feedback and suggestions for improving our manuscript. We kindly request your generous consideration of these points in the final evaluation of our paper.

Sincerely,

Authors of Submission 2649

---

### Decision · Program_Chairs · 2024-09-25

**Decision:**

Accept (poster)

**Comment:**

The paper proposes BELM, a generic formulation of exact inversion samplers. This includes existing exact inversion samplers and the analysis of the LTE allows to optimize the method. The reviewers appreciated the novelty of the framework, the theoretical analysis of the LTE as well as the practical potential of BELM. The rebuttal clarified some concerns (difference with existing classifier-free-text-guided image editing works) and provided additional experiments which were appreciated by the reviewers.

All the reviewers recommend acceptance, and I concur with this evaluation.